# FABP4-mediated lipid accumulation and lipolysis in tumor-associated macrophages promote breast cancer metastasis

Matthew Yorek[1†], Xingshan Jiang[1†], Shanshan Liu[1], Jiaqing Hao[1], Jianyu Yu[1], Anthony Avellino[1], Zhanxu Liu[2], Melissa Curry[3], Henry Keen[4], Jianqiang Shao[5], Anand Kanagasabapathy[1], Maying Kong[2], Yiqin Xiong[1], Edward R Sauter[6], Sonia L Sugg[7], Bing Li[1]*

[1]Department of Pathology, Holden Comprehensive Cancer Center, University of Iowa, Iowa City, United States; [2]Department of Bioinformatics and Biostatistics, University of Louisville, Louisville, United States; [3]Holden Comprehensive Cancer Center, University of Iowa Hospitals and Clinics, Iowa City, United States; [4]Iowa Institute of Human Genetics, University of Iowa, Iowa City, United States; [5]Central Microscopy Research Facility, University of Iowa, Iowa City, United States; [6]Division of Cancer Prevention, NIH/NCI, Rockville, United States; [7]Department of Surgery, University of Iowa, Iowa City, United States

*For correspondence:
bing-li@uiowa.edu

[†]These authors contributed equally to this work

Competing interest: The authors declare that no competing interests exist.

## eLife Assessment

This **important** paper uses elegant models, including genetic knock outs, to demonstrate that FABP4 contributes to lipid accumulation in tumor-associated macrophages, which seems to increase breast cancer migration. While the work is of high interest, the strength of the evidence relating to some of the conclusions is **incomplete** and the paper would benefit from some refinement. The work will be of interest to those researchers trying to link metabolism, the immune system, and cancer.

**Abstract** A high density of tumor-associated macrophages (TAMs) is associated with poorer prognosis and survival in breast cancer patients. Recent studies have shown that lipid accumulation in TAMs can promote tumor growth and metastasis in various models. However, the specific molecular mechanisms that drive lipid accumulation and tumor progression in TAMs remain largely unknown. Herein, we demonstrated that unsaturated fatty acids (FAs), unlike saturated ones, are more likely to form lipid droplets in murine macrophages. Specifically, unsaturated FAs, including linoleic acids (LA), activate the FABP4/CEBPα pathway, leading to triglyceride synthesis and lipid droplet formation. Furthermore, FABP4 enhances lipolysis and FA utilization by breast cancer cell lines, which promotes cancer cell migration in vitro and metastasis in vivo. Notably, a deficiency of FABP4 in murine macrophages significantly reduces LA-induced lipid metabolism. Therefore, our findings suggest FABP4 as a crucial lipid messenger that facilitates unsaturated FA-mediated lipid accumulation and lipolysis in TAMs, thus contributing to the metastasis of breast cancer.

## Introduction

As a main arm of innate immunity, macrophages are present in almost all tissues of an organism, where they play pivotal roles in maintaining tissue homeostasis or contributing to disease pathogenesis

(*Jin et al., 2021a*). In breast cancer, tumor-associated macrophages (TAMs) are the most abundant immune cells in the tumor microenvironment (TME), actively participating in tumor progression and metastasis (*Pollard, 2004*; *Condeelis and Pollard, 2006*; *Allison et al., 2023*). A high presence of TAMs in the TME correlates with a poorer prognosis of cancer patients in clinical studies (*Finak et al., 2008*; *Beck et al., 2009*; *Mehta et al., 2021*). Moreover, macrophage deletion via genetic or therapeutic approaches results in inhibition and regression of mammary tumors in pre-clinical studies (*Luo et al., 2006*; *Galmbacher et al., 2010*; *Linde et al., 2018*). These ample lines of evidence corroborate a crucial role for macrophages in facilitating breast cancer progression.

Macrophages are known for their heterogeneity regarding origin, phenotypes, and functional states (*Gordon and Taylor, 2005*). In response to diverse environmental stimuli from the environment, macrophages are believed to acquire a spectrum of functional states in vivo (*Stout and Suttles, 2004*; *Xue et al., 2014*). To understand their functional versatility, macrophages are commonly divided into the M1/M2 paradigm based on the concept of adaptive Th1/Th2 polarization (*Martinez et al., 2009*; *Mosser and Edwards, 2008*). Generally, Th1-derived cytokines polarize macrophages to M1 phenotype, which express inducible nitric oxide synthase and exhibit anti-tumor activities. Th2-cytokines polarized M2 macrophages highly express arginase 1, CD206, VEGFs, IL-10, TGFβ, *etc.*, supporting angiogenesis and favoring tumor growth (*Mantovani and Sica, 2010*; *Sica et al., 2002*). However, given the complex functional states of macrophages in vivo (*Murray et al., 2014*), the M1/M2 classification is likely oversimplified. Many studies demonstrate that TAMs do not follow the two extreme M1/M2 activation states in vivo (*Hao et al., 2018a*; *Weiss et al., 2018*). Considering that various extrinsic and intrinsic factors in the TME can cooperatively shape TAM phenotypes and functions (*DeNardo et al., 2009*; *Rolny et al., 2011*), identification of these factors is essential for targeting TAM-mediated cancer immunotherapy.

There is a growing realization that macrophage metabolism determines their functional outcome (*Ip et al., 2017*; *O'Neill and Pearce, 2016*). Several recent studies in multiple cancer models, including breast cancer (*Huggins et al., 2021*; *Masetti et al., 2022*; *Luo et al., 2020*; *Wu et al., 2019*), reported that lipid accumulation in TAMs promotes tumor growth and metastasis. Specifically, lipid accumulation promoted the expression of Arg1, VEGFα, MMP9, IL-6, PD-L1, CCL-6, *etc.*, thus conferring TAMs with immune suppressive and pro-tumorigenic phenotypes. Lipid droplet formation was as a characteristic feature of the pro-tumor TAMs, but the underlying molecular mechanisms driving lipid droplet formation in TAMs remain to be determined.

Despite morphological and compositional differences, lipid droplets are all formed by a phospholipid monolayer enclosing a neutral lipid core (*Olzmann and Carvalho, 2019*). A key step of lipid droplet formation is to synthesize neutral triacylglycerol (TAG), which requires the esterification of activated fatty acids (FA) to diacylglycerol (DAG) (*Tauchi-Sato et al., 2002*). During this step, enzymes including glycerol-3-phosphate acyltransferase (GPAT), acyl-CoA:1-acyl-glycerol-3-phosphate acyltransferase (AGPAT), DAG acyltransferases (DGATs), are key in the biogenesis of neutral lipids. As such, factors engaging with FA transport and regulating key enzyme activity are deemed to be critical for lipid droplet formation in TAMs.

Due to the insolubility of hydrophobic FAs, fatty acid binding proteins (FABPs) have evolved to solubilize FAs, facilitating their transport and responses (*Furuhashi and Hotamisligil, 2008*). Among FABPs, adipose FABP (A-FABP or FABP4) is the most well studied given its striking biology in regulating macrophage lipid metabolism and functions in multiple disease settings (*Jin et al., 2021a*; *Li et al., 2020*). Our pre-clinical studies demonstrated that a subset of macrophages with the phenotype of CD11b[+]F4/80[+]Ly6C[-]CD11c[-]MHCII[-] highly express FABP4, facilitating macrophage lipid processing and patrolling functions (*Jin et al., 2021a*; *Zhang et al., 2017c*). Moreover, FABP4 was identified as a functional marker for protumor TAMs by enhancing the oncogenic IL-6/STAT3 signaling (*Hao et al., 2018a*). Given the emerging role of lipid accumulation in protumor TAMs, we hypothesized that FABP4 might function as a key molecular sensor promoting lipid accumulation and utilization in TAMs.

Utilizing a combination of in vitro cellular studies with various species of FAs, in vivo macrophage single-cell sequencing in mouse models, and macrophage/FABP4 analysis in human breast cancer specimens, we demonstrate that FABP4 plays a pivotal role in mediating lipid droplet formation and subsequent lipolysis for tumor utilization, thus contributing to TAM-mediated tumor growth and metastasis in breast cancer.

## Results

### Unsaturated FAs promote lipid accumulation in macrophages

Hostile TME, especially in breast cancer, is characterized by enriched FAs and other lipid species, which foster immunosuppression and support tumor growth and metastasis (*Xu et al., 2021*; *Zhang et al., 2017b*). Since palmitic acid (PA, 16:0), stearic acid (SA, 18:0), oleic acid (OA, 18:1), and linoleic acid (LA, 18:2) are the most common FAs in humans, as shown in *Figure 1A*, we treated macrophages with individual albumin-conjugated FAs and determined their fate. Using a macrophage cell line, we demonstrated that all the FAs studied were oxidized by mitochondria as evidenced by the increased oxygen consumption rate (OCR) when compared to the bovine serum albumin (BSA) control (*Figure 1B*). Interestingly, when measuring their impact on other cellular organelles, FAs exhibited dichotomous effects, with saturated FAs increasing lysosome contents (*Figure 1C*), while unsaturated FAs promoting ER (*Figure 1D*) and, more dramatically, lipid droplet (LD) formation (*Figure 1E*) in macrophages. Using primary peritoneal macrophages (*Figure 1—figure supplement 1A*) and bone-marrow-derived macrophages (*Figure 1—figure supplement 1B*), we observed the similar phenomenon that unlike saturated FA, unsaturated FA significantly induced LD formation in these macrophages. Oil Red O staining confirmed the presence of LD formation (*Figure 1—figure supplement 1C*). Of note, the observation of unsaturated FA-induced LD formation in macrophages was consistent regardless of FA concentrations (*Figure 1—figure supplement 1D*) and treatment durations (*Figure 1—figure supplement 1E*). With a multispectral imaging analysis, we demonstrated that OA/LA-induced LD accumulation was not associated with the lysosome, but colocalized in the ER (*Figure 1F*). Transmission electron microscope further confirmed that OA and LA specifically induced LD formation in the cytoplasm (*Figure 1G*), which were bud from ER (*Figure 1—figure supplement 1F*). Collectively, these data suggest that unsaturated FA-induced LD biogenesis in macrophages mainly occurs in the ER.

### Unsaturated FAs upregulate key enzymes for LD biogenesis

It is well known that key enzymes, including GPATs, AGPATs, LIPINs, DGATs, are involved in TAG synthesis and LD biogenesis (*Figure 2A*). To determine how unsaturated FAs, but not saturated FAs, induced LD biogenesis in macrophages, we treated macrophages with either LA or PA, and measured their impact on LD biogenesis-related enzymes. In response to PA or LA treatment, *Acs1*, *Cpt1a*, *Cpt1b*, *HMGCR*, *Acat1*, and *Acat2* had similar expression pattern in macrophages (*Figure 2—figure supplement 1A–F*), suggesting that both FA types exerted similar effects on FA activation, mitochondrial oxidation, cholesterol synthesis and esterification. Interestingly, genes encoding rate-limiting enzymes in TAG synthesis, including *Gpam1*, *Dgat1* and *Dgat2*, were significantly increased by treatment with LA, but not PA (*Figure 2B–H*), suggesting a unique role of LA in transcriptional upregulation of neutral lipid synthesis pathways. We observed by confocal microscopy that macrophages treated with LA, but not BSA or PA, exhibited elevated levels of LipidTOX, GPAT1, and DGAT1 (*Figure 2I*). Quantitative measurements showed a significant increase of lipid accumulation (LipidTOX), GPAT1 and DGAT1 proteins in the LA-treated vs. the BSA or PA groups (*Figure 2J–M*). Moreover, intracellular flow cytometric staining (*Figure 2—figure supplement 1G–I*) and quantification by mean fluorescent intensity (MFI) (*Figure 2—figure supplement 1J–L*) also confirmed our observations that unsaturated FAs significantly increased lipid accumulation by enhancing GPAT/DGAT-mediated lipid accumulation in macrophages.

### C/EBPα transcriptionally controls LA-induced TAG synthesis

To determine how unsaturated LA transcriptionally upregulated gene expression in the TAG synthesis pathway, we searched for transcription factor binding sites, focusing on the pooled genes related to TAG synthesis pathways. Two transcriptional factors, C/EBPα and C/EBPβ, were predicted to commonly bind the promotor regions of these genes (*Figure 3—figure supplement 1A*). When macrophages were treated with PA or LA, C/EBPα (*Figure 3A*), but not C/EBPβ (*Figure 3—figure supplement 1B*), was significantly upregulated by LA treatment. Confocal microscopy showed that C/EBPα was mainly localized in the cytosol with BSA and PA treatment but was significantly upregulated in the nuclei with LA treatment (*Figure 3B and C*). In contrast, C/EBPβ was present in both cytosol and nuclei regardless of PA or LA treatment (*Figure 3—figure supplement 1C and D*). These results suggest that LA treatment induces specific activation of C/EBPα in macrophages.

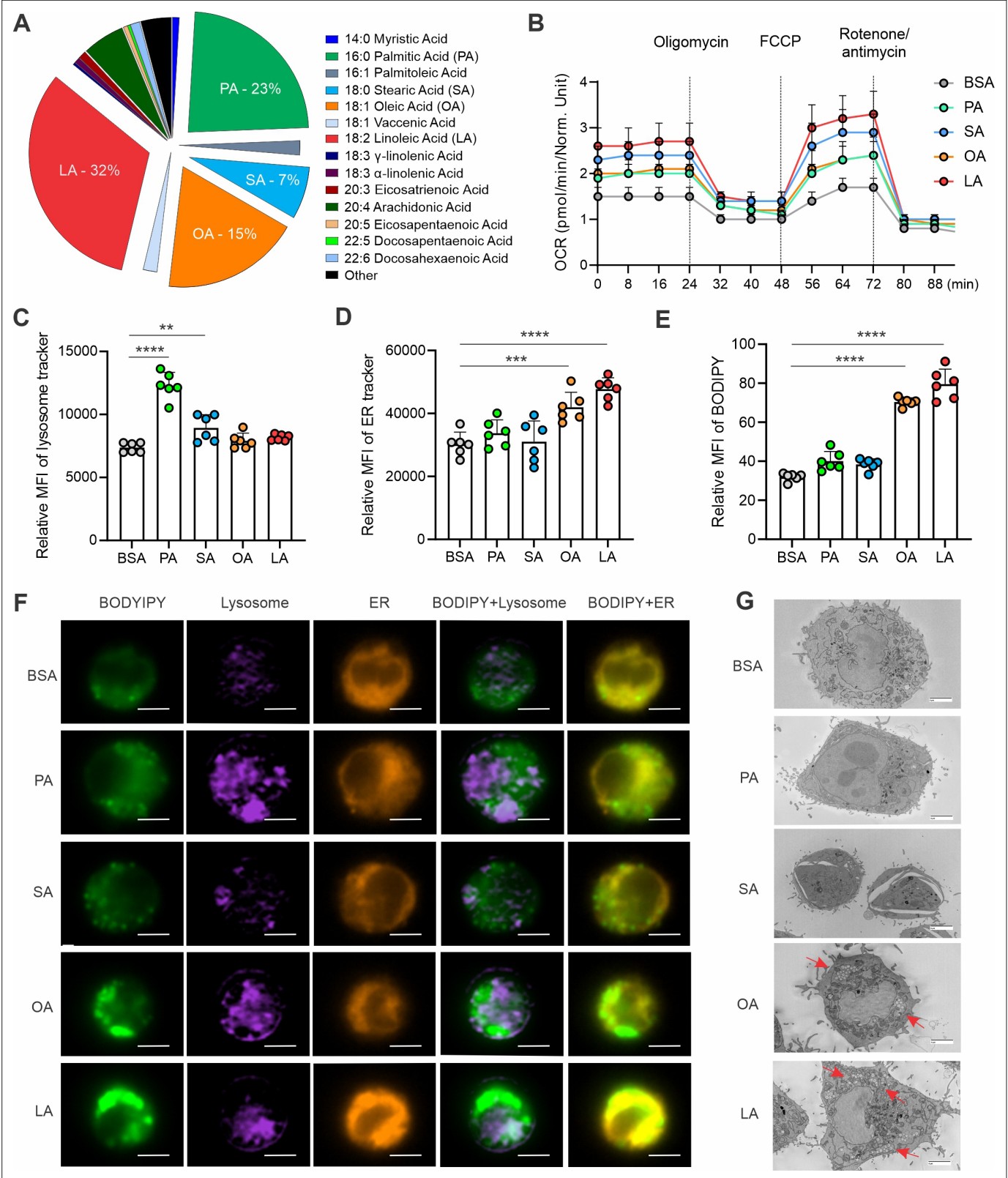

**Figure 1.** Unsaturated FAs form lipid accumulation in macrophages. (**A**) Pie chart showing main compositions of free fatty acids in the serum of healthy humans. (**B**) Measurement of oxygen consumption rate (OCR) in macrophages treated with 200 µM of PA, SA, OA, LA or control BSA, respectively, under basal conditions or following the addition of oligomycin, FCCP or the electron transport inhibitor Rotenone/antimycin by a seahorse XF-96 analyzer (n=5). (**C–E**) Macrophages were treated with 200 µM of PA, SA, OA, LA or BSA for 4 hr. Flow cytometric analysis of lysosome, ER and lipid droplet

*Figure 1 continued on next page*

*Figure 1 continued*

formation by measuring mean fluorescent intensity (MFI) of lysosome tracker (**C**), ER tracker (**D**) and BODIPY (**E**) in macrophages. (**F**) Multispectral imaging analysis of BODIPY (green), Lysosome (purple), ER (orange), and merged images showing the colocalization status of BODIPY/lyso-tracker and BODIPY/ER-tracker in macrophages treated with PA, SA, OA, LA, or BSA at 200 µM for 4 hr. Scale bar: 7µM. (**G**) Analysis of lipid droplet formation (red arrow) in macrophages treated with BSA, PA, SA, OA, and LA by transmission electron microscope. Data are shown as mean ± SD in panel B-E (** p≤0.01, *** p≤0.001, **** p≤0.0001, as compared to the control BSA group, unpaired Student t test).

The online version of this article includes the following figure supplement(s) for figure 1:

**Figure supplement 1.** Unsaturated FA form lipid droplets in different macrophages.

To further dissect the role of C/EBPs in LA-treated macrophages, we knocked down C/EBPα (*Figure 3D*) and C/EBPβ (*Figure 3—figure supplement 1E*), respectively, in macrophages. LA-induced gene expression of *Gpam1, Dgat1, Dgat2* was abrogated when C/EBPα was silenced (*Figure 3E–G*). Moreover, the uptake of FA (*Cd36*), lysosome lipase (*Lipa*) and mitochondrial oxidation (*Cpt1b*) were significantly increased in C/EBPα-knockdown macrophages (*Figure 3H–J*), suggesting that C/EBPα knockdown not only inhibited lipid synthesis, but also promoted lipolysis and oxidation. As such, C/EBPα knockdown significantly inhibited LA-induced neutral lipid accumulation as shown by the staining of BODIPY (*Figure 3K*) and Oil Red O (*Figure 3L and M*) in macrophages. By contract, C/EBPβ silencing did not affect the expression of *Gpam1, Cd36*, and *Lipa* expression (*Figure 3—figure supplement 1F–H*). Although C/EBPβ knockdown appeared to inhibit the expression of *Dgat1* and *Dgat2*, it also inhibited *Cpt1b*-mediated FA oxidation (*Figure 3—figure supplement 1I–K*). Overall, C/EBPβ knockdown resulted in lipid accumulation in response to LA treatment in macrophages (*Figure 3—figure supplement 1L*). Altogether, these data suggest that C/EBPα, but not C/EBPβ, is a critical transcriptional factor controlling LA-induced TAG synthesis and LD formation in macrophages.

## FABP4 mediates LA-induced C/EBPα activation

Given the insolubility of long chain FAs, it was of great interest to dissect how LA, unlike SA, specifically activated C/EBPα in macrophages. As FA chaperones, FABPs bind different FAs, facilitating their transport and responses. We measured the profile of common FABP members in macrophages and demonstrated that FA treatment mainly induced expression of FABP4, but not FABP3 and FABP5, in macrophages (*Figure 4A–C*). Analyzing the TCGA human breast cancer database, we found that compared to other FABP family members FABP4 was mostly correlated with C/EPBα (*Figure 4—figure supplement 1A and B*), but not with other C/EBP members (*Figure 4—figure supplement 1C*). The positive associations of FABP4 with C/EBPα and key TAG synthesis genes were also evidenced in multiple GEO datasets (*Figure 4—figure supplement 1D–F*).

To gain insights into our observations, we performed single cell RNA-sequencing using splenic macrophages. UMAP plots showed that macrophages consisted of 13 clusters based on untargeted gene expression profiles (*Figure 4—figure supplement 1G*). FABP4 was enriched in clusters 0, 1, and 2 (*Figure 4D*). Compared to other non-enriched clusters (*Fabp4-*), GO molecular function analysis showed that FABP4 was mainly involved in protein-lipid complex, lipid binding and signaling receptors (*Figure 4—figure supplement 1H*). Moreover, *Fabp4* enriched clusters (*Fabp4+*) were highly associated with genes known to be critical for lipid uptake and accumulation, including *Cd36, PPARγ, CEBPA, Plin2* (*Figure 4E–I*). In contrast, *Fabp4* enrichment was not related to *Fabp5, Acsl1, Lipa, Fasn, and Cpt1b* (*Figure 4J, Figure 4—figure supplement 1I–L*), suggesting a unique role of FABP4 in lipid uptake and storage in macrophages.

Confocal microscopy showed that FABP4 was mainly expressed in the cytosol in BSA- or PA-treated cells, whereas LA treatment led to FABP4 present both in the cytosol and nuclei (*Figure 4K*), suggesting a LA-induced FABP4 nuclear translocation effect for potential nuclear transactivation. To confirm the role of FABP4, we measured *C/EBPα* expression in FABP4-deficienct macrophages. Deletion of FABP4 in macrophages abrogated LA-induced C/EBPα expression in nuclei (*Figure 4L and M*). We further confirmed that FABP4 deficiency significantly blocked LA-induced expression of C/EBPα (*Figure 4—figure supplement 1M and N*), but not C/EBPβ (*Figure 4—figure supplement 1O and P*), in primary peritoneal macrophages and bone-marrow-derived macrophages (BMMs). Collectively, these data demonstrated a pivotal role for FABP4 in mediating LA-induced C/EBPα activation in macrophages.

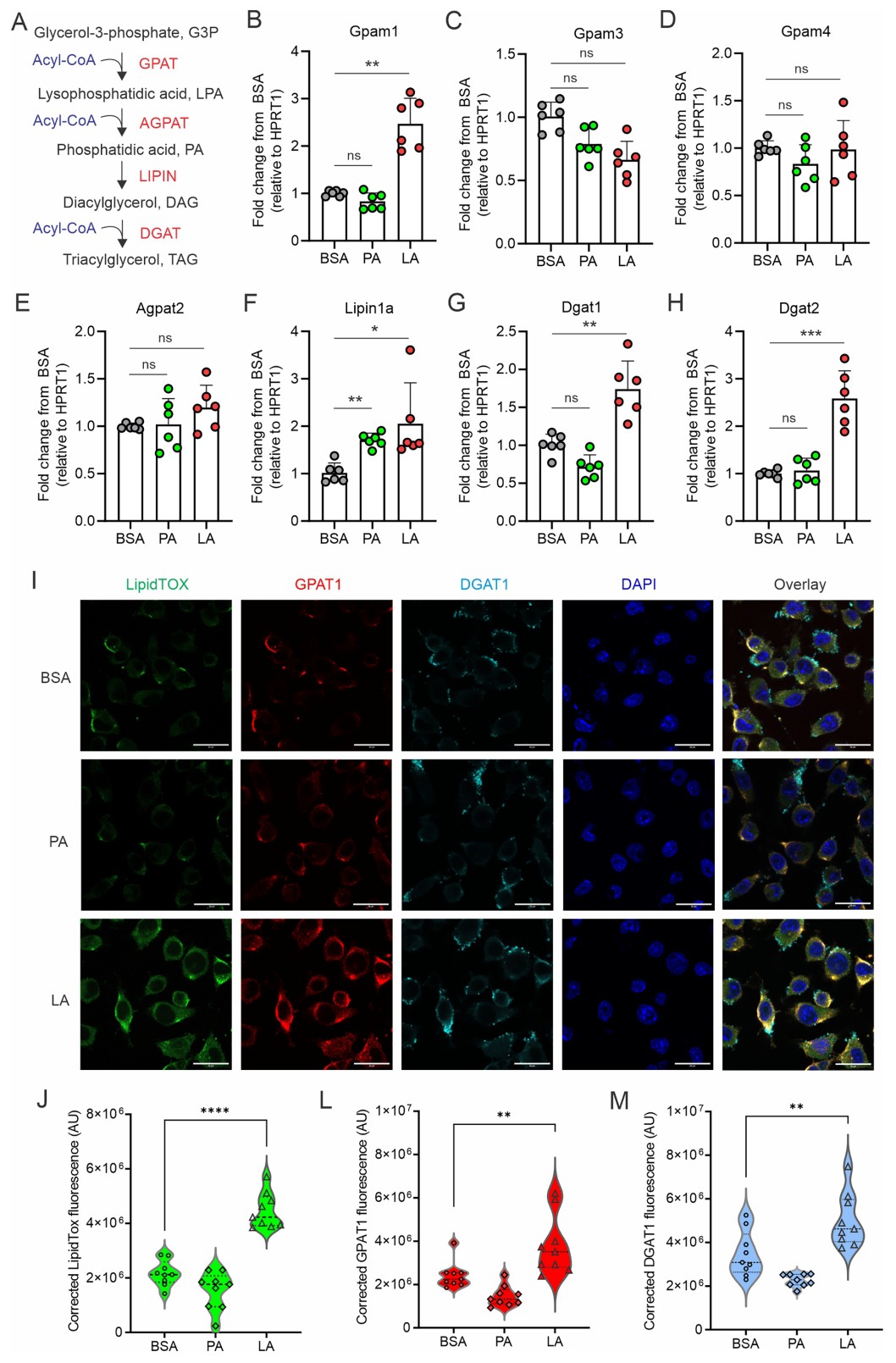

**Figure 2.** LA induces the expression of key enzymes of triacylglycerol synthesis in macrophages. (**A**) Key enzymes in the biosynthesis of triacylglycerol. (**B–H**) Analyzing the expression of genes encoding key enzymes, including Gpam1 (**B**), Gpam3 (**C**), Gpam4 (**D**), Agpat2 (**E**), Lipin1a (**F**), Dgat1 (**G**), Dgat2 (**H**), in the triacylglycerol biosynthesis pathway in macrophages treated with 400 μM of PA, LA or BSA for 4 hr by real-time PCR. (**I**) Representative

*Figure 2 continued on next page*

*Figure 2 continued*

confocal images of lipid accumulation by LipidTOX staining (green), expression of GPAT1 (red), DGAT1 in macrophages treated with BSA, PA, and LA (400 µM) overnight. (**J–M**) Quantification of lipid accumulation (**J**), protein levels of GAPT1 (**K**) and DGAT1 (**M**) in macrophages treated with BSA, PA, or LA (400 µM) overnight. Data are shown as mean ± SD in panel B-H, J-M (** p≤0.01, *** p≤0.001, **** p≤0.0001, ns, non-significant, as compared to the control BSA group, unpaired Student t test).

The online version of this article includes the following figure supplement(s) for figure 2:

**Figure supplement 1.** LA induces key enzyme expression in triglyceride biosynthesis.

## FABP4 expression enhances lipid accumulation in macrophages

Given the critical role of FABP4 in activating the LA-C/EBPα axis, we reasoned that FABP4 deficiency suppressed LA-induced lipid accumulation in macrophages. To this end, we first measured if FABP4 deficiency reduced key enzymes of TAG synthesis in different macrophages. Using FABP4 wild type (WT) and knockout (KO) macrophage cell lines (*Figure 5A*), we demonstrated that upregulation of *Gpam1*, *Dgat1*, and *Dgat2*, but not *Gpam4* and *Agpat2*, in response to LA treatment in WT macrophages, was significantly reduced when FABP4 was genetically depleted (*Figure 5B–F*). Similarly, using peritoneal macrophages purified from WT and KO mice, we confirmed that *FABP4* deficiency significantly reduced the expression of LA-induced *Gpam1, Dgat1, Dgat2* (*Figure 5—figure supplement 1A–D*). Of note, FABP4 deficiency also reduced expression of other lipid transporters, including *Cd36* and *Fabp5* (*Figure 5—figure supplement 1E–F*), suggesting a general inhibition of fatty acid uptake and metabolism in the absence of FABP4 in primary macrophages.

To evaluate the role of FABP4 in lipid accumulation, we measured GPAT1 and DGAT1 protein expression and lipid droplet formation in LA-treated WT and KO macrophages. Confocal microscopy showed that GPAT1, DGAT1, and lipidTOX were colocalized in the cytoplasm of WT macrophages whereas FABP4 deficiency reduced the expression of GPAT1, DGAT1, and lipidTOX in KO macrophages (*Figure 5G*). Quantitative analysis demonstrated that GPAT1, DGAT1, and lipidTOX levels were significantly reduced when FABP4 was absent (*Figure 5H–J*). Using transmission electron microscopy, we further confirmed that FABP4 deficiency compromised LA- and OA-induced lipid accumulation in macrophages (*Figure 5L*, *Figure 5—figure supplement 1G*). The role of FABP4 in lipid accumulation was corroborated by analysis of primary peritoneal macrophages from FABP4 WT and KO mice. As shown in *Figure 5—figure supplement 1H and I*, LA-induced neutral lipid accumulation in macrophages was significantly reduced when FABP4 was deficient. Collectively, these data suggest that FABP4 plays a pivotal role in TAG synthesis and neutral lipid accumulation in macrophages.

## Macrophage expression of FABP4 promotes breast cancer migration

To further dissect the functionality of FABP4 in TAMs, we analyzed FABP4-associated gene pathways in the TCGA breast cancer datasets. Interestingly, FABP4 expression was highly associated with the lipolysis regulation pathway, including genes of β-adrenergic receptors (*Adrb2*), G-protein-coupled receptor signaling (*Adcy4*), and lipolysis enzymes (*Pnpla2, Lipe*) (*Figure 6—figure supplement 1A–D*). When FABP4 WT and KO macrophages were treated with PA or LA, FABP4 deficiency impaired LA-induced lipolysis in macrophages as evidenced by significantly reduced levels of *Adrb2*, *Adcy4*, *Pnpla2*, and *Lipe* (*Figure 6A–D*), suggesting a critical role of FABP4 in mediating lipolysis-associated pathways in macrophages. Given the protumor role of lipid accumulation in TAMs, we treated FABP4 WT or KO macrophages with PA, LA, or BSA, and evaluated their protumor function using trans-well tumor migration assays (*Figure 6E*). As shown in *Figure 6F and G*, compared to BSA- or PA-treated WT macrophages, LA-treated macrophages greatly promoted MDA-MB-231 cell migration. Notably, FABP4 deficiency significantly reduced LA-mediated tumor migration. Using E0771 mammary tumor cells, we confirmed the essential role of FABP4 in mediating LA-induced tumor migration effect in macrophages (*Figure 6H and I*). The FABP4/LA-induced tumor migration effect was not due to tumor proliferation as Ki67 expression was similar among different FA-treated groups (*Figure 6—figure supplement 1E*). Instead, lipid droplets formed by LA in WT macrophages were depleted following coculture with tumor cells (*Figure 6J*), showing the pro-migration effect of FABP4-mediated lipolysis. Furthermore, we collected the conditioned media collected from FA-treated macrophages and demonstrated that only media from LA-treated macrophages promoted migration

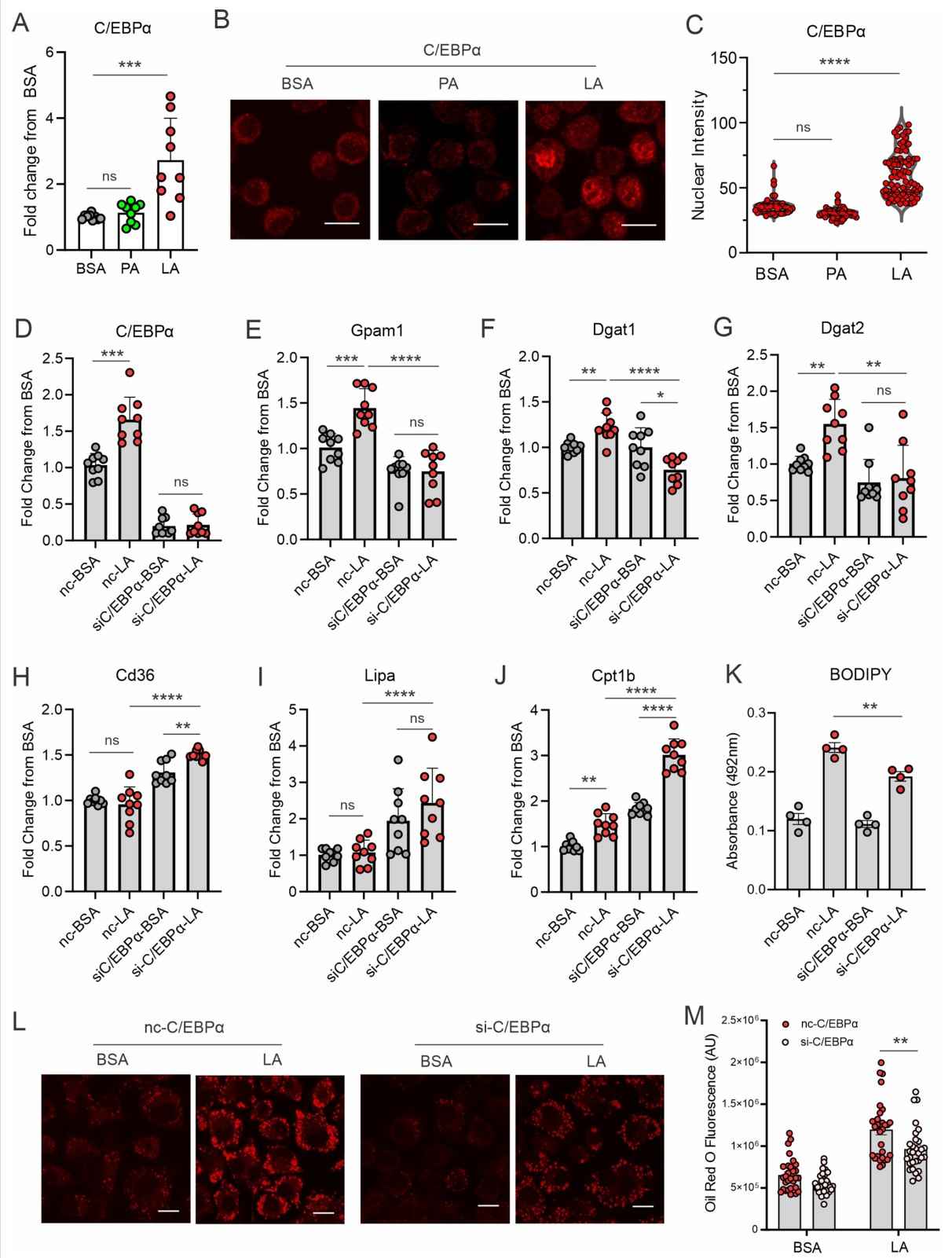

**Figure 3.** LA induces lipid accumulation through activating the C/EBPα pathway. (**A**) Measurement of C/EBPα gene expression levels in macrophages treated with BSA, PA, or LA (400 μM) for 4 hr by real-time PCR. (**B**) Representative confocal images of C/EBPα protein expression (red) in macrophages treated with BSA, PA, or LA overnight. (**C**) Quantification of C/EBPα nuclear expression in macrophages treated with BSA, PA, or LA overnight by Image J analysis. (**D–J**) Real-time PCR analysis of the levels of C/EBPα (**D**), Gpam1 (**E**), Dgat1 (**F**), Dgat2 (**G**), Cd36 (**H**), Lipa (**I**), and Cpt1b (**J**) in macrophages

*Figure 3 continued on next page*

*Figure 3 continued*

transfected with 40 nM C/EBPα siRNA or control siNC and then treated with BSA or LA for 4 hr. (**K**) Measurement of BODIPY fluorescent intensity in C/EBPα-silencing or control macrophages treated with BSA or LA using a fluorescence spectroscopy. (**L**) Representative confocal images of Oil Red O staining in C/EBPα-silencing or control macrophages treated with BSA or LA (bar, 10 μM). (**M**) Quantification of Oil Red O fluorescence intensity in C/EBPα-silencing or control macrophages treated with BSA or LA. Data are shown as mean ± SD in panel A, C-K and M (*p≤0.05, ** p≤0.01, *** p≤0.001, **** p≤0.0001, ns, non-significant, as compared to the control BSA group or control siNC group, unpaired Student t test).

The online version of this article includes the following figure supplement(s) for figure 3:

**Figure supplement 1.** LA-induced lipid accumulation in macrophages was undependable on CEBPβ.

of MDA-MB-231 and E0771 in a FABP4-dependent manner (*Figure 6—figure supplement 1F–I*). These data suggested that intracellular FABP4 was secreted from macrophages into the extracellular environment to mediate the pro-migration effect. Indeed, LA-treated macrophages exhibited the highest levels of FABP4 compared to other groups (*Figure 6—figure supplement 1J*). Altogether, FABP4 expression in macrophages was essential for LA-induced pro-tumor effects.

## High expression of FABP4 in TAMs is associated with breast cancer metastasis

Recently, meta-analysis demonstrated that a high density of TAMs, especially CD163+ TAMs, predicted poor survival outcomes in breast cancer (*Allison et al., 2023*). When we analyzed pro-tumor activity of TAMs using human breast cancer tissue, we verified that high expression of the macrophage marker CD163 was significantly associated with a reduced overall survival of these breast cancer patients in multiple breast cancer datasets (*Figure 7—figure supplement 1A–1D*). To verify the results from these publicly accessible datasets, we collected breast cancer tissue specimens from a cohort of 59 women with different subtypes of breast cancer (*Supplementary file 1*) and performed immunohistochemical (IHC) staining. Interestingly, CD163+ TAMs were positively correlated with breast tumor size (*Figure 7A and B*), corroborating the pro-tumor function of CD163+ macrophages in breast cancer. We also noticed that compared to lean patients, obese patients had more CD163+ macrophages in the TME (*Figure 7—figure supplement 1E and F*), suggesting that dysregulated lipids in obese patients promoted macrophage tumor infiltration. Given the pivotal role of FABP4 in lipid metabolism in macrophages, we analyzed FABP4 expression in CD163+ TAMs. Interestingly, FABP4 expression significantly correlated with CD163+ macrophages (p<0.001; *Figure 7C*). Among the cohort, 22% of patients exhibited lymph node and distant metastasis (*Figure 7D*). When we compared macrophages and FABP4 expression in tumor tissues in patients with or without tumor metastasis (*Figure 7E*), we showed that tumors from metastatic patients had higher expression of CD163+ macrophages (*Figure 7F*) and FABP4 expression (*Figure 7G*). We also examined the association of other variables, including age, tumor grade, multifocality, lymphovascular invasion, mortality, hormone status, with CD163 and FABP4 expression in tumor tissues. Higher tumor grade and mortality were associated with more FABP4 expression in macrophages (*Figure 7H*, *Supplementary file 2*). Moreover, triple negative breast cancer exhibited higher FABP4 expression in macrophages than ER+ or HER2+ breast cancer (*Figure 7—figure supplement 1G and H*). Collectively, CD163+ TAMs, especially those with FABP4 expression, are positively associated with tumor growth and metastasis in breast cancer patients.

## Discussion

Breast cancer is the most prevalent cancer in women globally (*Arnold et al., 2022*). The World Health Organization reported in 2020 that 2.3 million women were diagnosed with breast cancer, with 685,000 deaths from the disease (*Arnold et al., 2022*; *Forouzanfar et al., 2011*). Given that metastatic breast cancer is the leading cause of breast cancer-related death (*Valastyan and Weinberg, 2011*), identifying new cellular and molecular mechanisms underlying breast cancer metastasis is an urgent need for breast cancer research. The TME of breast cancer is lipid enriched, and these lipids can be taken up by macrophages, leading to their alternative activation (*Huang et al., 2014*). Alternative activated macrophages are known to stimulate angiogenesis and promote tumor progression (*Mantovani and Sica, 2010*). However, the molecular mechanisms by which extracellular lipids contribute to intracellular lipid accumulation and pro-tumor functions in macrophages are largely unexplored.

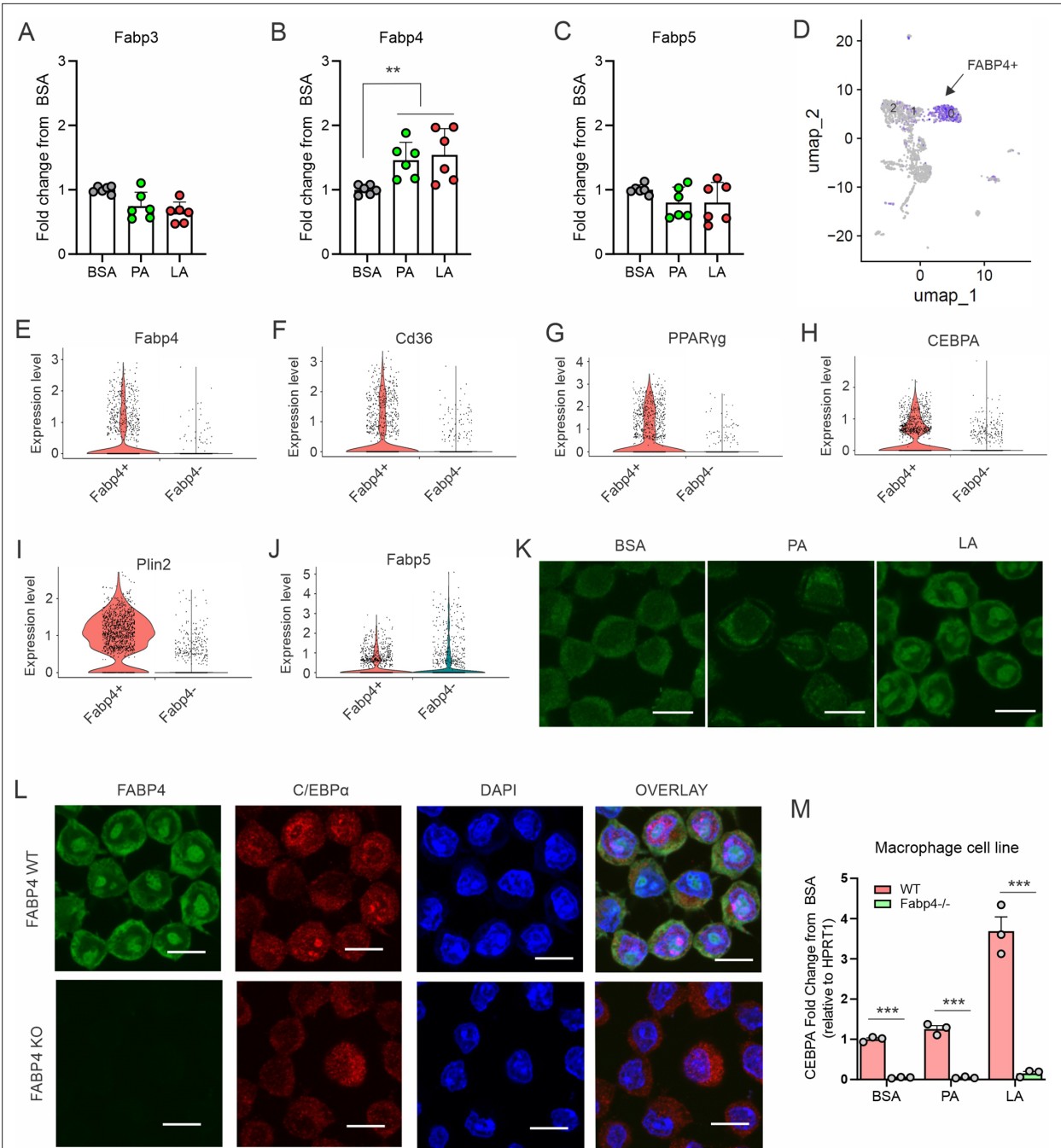

**Figure 4.** FABP4 mediates LA-induced C/EBPα expression in macrophages. (**A–C**) Analysis of the expression of FABP family members, including Fabp3 (**A**), Fabp4 (**B**) and Fabp5 (**C**), in macrophages treated with BSA, PA or LA (400 μM) for 4 hours. (**D**) UMAP of FABP4-positive macrophage subsets using mouse spleen single-cell RNA sequence analysis. (**E–J**) Violin plots showing relative expression levels of genes, including Fabp4(E), Cd36 (**F**), PPARγ (**G**), CEBPA (**H**), Plin2 (**I**) and Fabp5 (**J**) between Fabp4 +vs Fabp4- macrophages indicated in (**C**). (**K**) Confocal analysis of FABP4 expression in macrophages treated with BSA, PA or LA (400 μM) (bar, 10 μM). (**L**) Comparison of the expression of FABP4 and C/EBPα between FABP4 wildtype (WT) and knockout (KO) macrophages in response to LA treatment (400 μM) (bar, 10 μM). (**M**) Realtime PCR analysis of CEBPA expression in WT and FABP4-/- macrophages treated with BSA, PA, and LA (400 μM). Data are shown as mean ± SD in panel A and L (** p≤0.01, *** p≤0.001, as compared to the control BSA group or FABP4-/- group, unpaired Student t test).

The online version of this article includes the following figure supplement(s) for figure 4:

**Figure supplement 1.** FABP4 mediates LA-induced CEBPα, but not CEBPβ, activation.

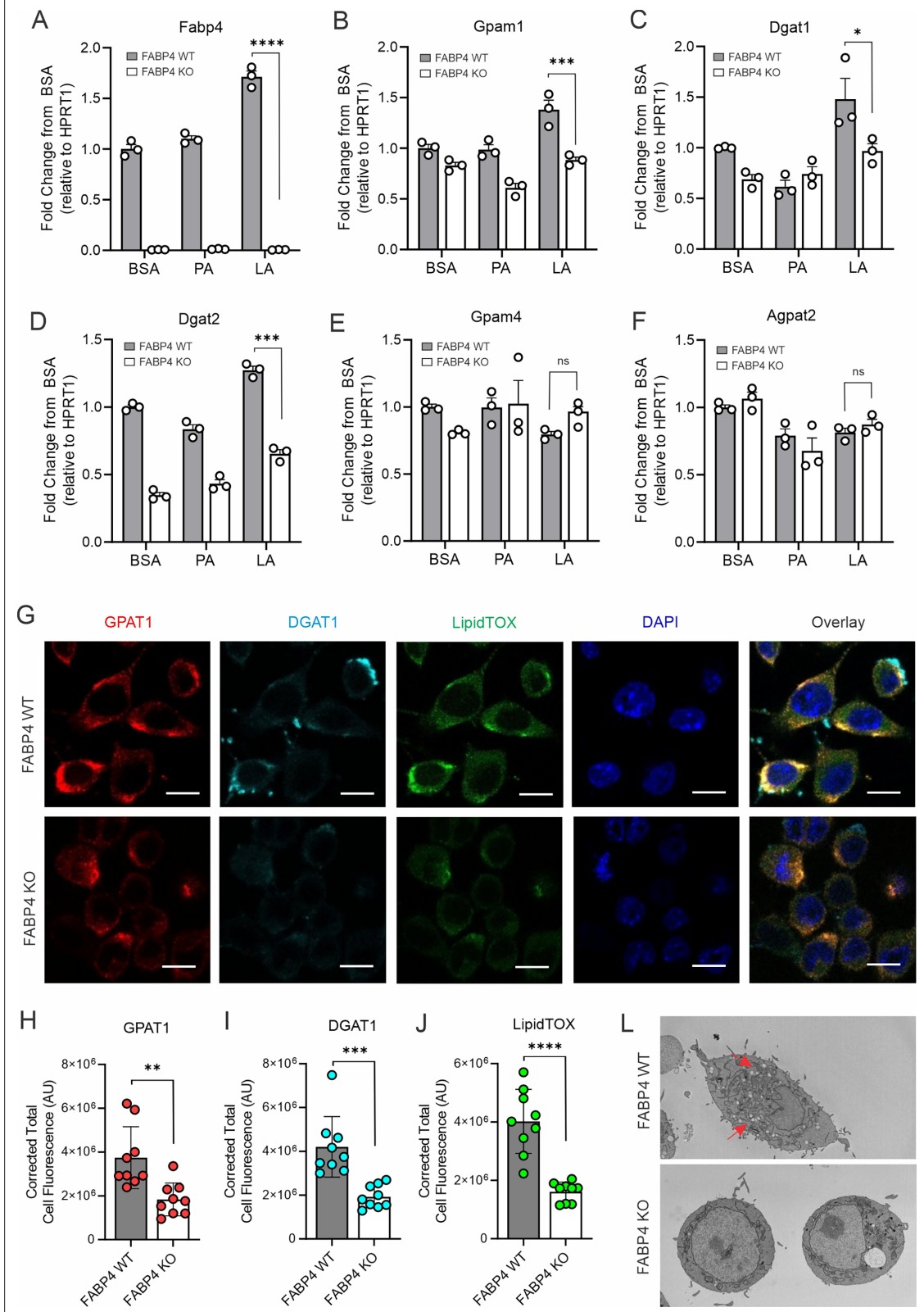

**Figure 5.** FABP4 deficiency reduces LA-induced lipid accumulation in macrophages. (**A–F**) real-time PCR analysis of FABP4 (**A**) and genes encoding key enzymes for triglycerol biosynthesis, including Gpam1 (**B**), Dgat1 (**C**), Dgat2 (**D**), Gpam4 (**E**), Agpat2 (**F**) in WT and FABP4 KO macrophages treated with BSA, PA, or LA (400 µM). (**G**) Confocal analysis of protein expression of GPAT1 (red), DGAT1 (cyan) and lipid accumulation (LipidTOX staining, green) in LA-treated WT and FABP4 KO macrophages (bar, 10 µM). (**H–J**) Expression levels of GPAT1 (**H**), DGAT1 (**I**) and LipidTOX (**J**) as indicated in panel G

*Figure 5 continued on next page*

*Figure 5 continued*

were quantified by Image J. (**L**) Flow cytometric analysis of neutral lipid accumulation as shown by BODIPY staining in WT and FABP4 KO macrophages treated with BSA or LA. (**M**) Transmission electron microscope showing lipid droplet staining in WT and FABP4 KO macrophages treated with LA. Data are shown as mean ± SD in panels A-F, H-L (* p≤0.01, p** p≤0.01, *** p≤0.001, ns, non-significant as compared to the control BSA group or FABP4-/- group, unpaired Student t test).

The online version of this article includes the following figure supplement(s) for figure 5:

**Figure supplement 1.** Deficiency of FABP4 reduces lipid droplet formation in macrophages.

As professional phagocytes, macrophages are engaged with uptake of extracellular lipids to mitigate their toxicity and to process them for use by themselves or adjacent cells (*Batista-Gonzalez et al., 2019*). Macrophages have evolved to mainly express FABP4 and FABP5 to facilitate FA transport and metabolism. We previously demonstrated that heterogenous macrophages exhibit different FABP expression profiles and activities. For instance, FABP5 expression in skin macrophages facilitates high fat diet-induced IL-1 family cytokine responses (e.g. IL-1β, IL-36γ) while FABP4 expression in TAMs promotes oncogenic IL-6/STAT3 signaling (*Hao et al., 2018a*; *Hao et al., 2022*; *Zhang et al., 2015*). We further demonstrated that heterogenous macrophages exhibit a unique FABP expression profile. FABP4 is prominently expressed in CD11b+F4/80+CD11c-MHCII- macrophage subsets whereas FABP5 is mainly expressed in CD11b+F4/80+MHCII+CD11c+ macrophage subsets (*Hao et al., 2018a*; *Zhang et al., 2017c*; *Zeng et al., 2018*), suggesting a distinct role of individual FABP family members in macrophages (*Hao et al., 2018a*; *Zhang et al., 2014*). Interestingly, when macrophages were exposed to different species of dietary FAs, FABP4 appeared to be more responsive to FA treatment. Notably, FABP4 was mainly located in the cytoplasm under PA treatment but translocated to the nucleus in response to LA treatment. This was consistent with previous studies showing LA as an activating ligand for FABP4 nuclear translocation and PPARγ activation (*Gillilan et al., 2007*). Considering that PPARγ induces CEBPα expression by binding to its enhancer and they synergistically enhance lipid accumulation in adipocytes (*Lau et al., 2023*; *Madsen et al., 2014*), we demonstrated that LA treatment specifically upregulated CEPBα expression and induced lipid droplet formation in macrophages. Importantly, CEBPα transcriptionally upregulated key enzymes for TAG synthesis, including GPAT1 and DGATs. Moreover, silencing CEBPα expression significantly inhibited LA-induced GPAT1 and DGAT1 expression and lipid droplet accumulation. Therefore, our data suggest a previously underappreciated axis of LA/FABP4/CEBPα in mediating lipid accumulation in macrophages.

To determine the role of the accumulated lipids in macrophages, we noticed that FABP4 expression was not only associated with the LA/CEBPα/DGAT-mediated lipid accumulation pathway, but also highly correlated with the β-AR/ATGL/HSL-mediated lipolysis pathway in breast cancer. Genetic deletion of FABP4 inhibited both lipid accumulation and lipolysis pathways, confirming a critical role of FABP4 in lipid metabolism in macrophages. When breast cancer cells were cocultured with FA-exposed macrophages, LA-treated macrophages dramatically promoted breast cancer cell migration as compared to PA- or BSA-treated cells, suggesting a specific effect of LA-mediated lipid accumulation and utilization in macrophages. Moreover, FABP4 deficiency impaired LA-mediated protumor migration effect of macrophages, further supporting that FABP4-mediated lipolysis in macrophages was critical for lipid utilization by breast cancer cells.

TAMs are the most abundant immune cells in breast cancer. To interrogate TAM's role in the TME, M1/M2 dichotomy has been commonly proposed to polarize TAM functions. However, there have been no convincing markers or precise molecular mechanisms to determine TAM functions in vivo. By analyzing FABP4 expression in TAMs with human breast cancer specimens, we found that FABP4 expression in TAMs, especially CD163+ TAMs, correlated directly with tumor size, grade, metastasis and even patient survival. Considering the critical role of FABP4 in mediating lipid accumulation and lipolysis, it might function as a key player linking extracellular lipids to lipid storage and utilization in TAMs. Of note, compared to lipid-processing macrophages, most epithelial-cell-derived cancer cells do not express FABP4, thus FABP4 may serve as a functional marker for lipophilic pro-tumor TAMs.

It is worth noting that dietary fats are linked to both breast cancer incidence and survival in epidemiologic studies (*Gopinath et al., 2022*). However, the exact role of major dietary FAs, including PA, SA, OA, and LA, in breast cancer risk and progression remains unclear. Given the observations that saturated fat intake increases the risk of cardiovascular diseases, the American Dietary Guidelines recommend the consumption of unsaturated fats over saturated fats in our diets (https://www.

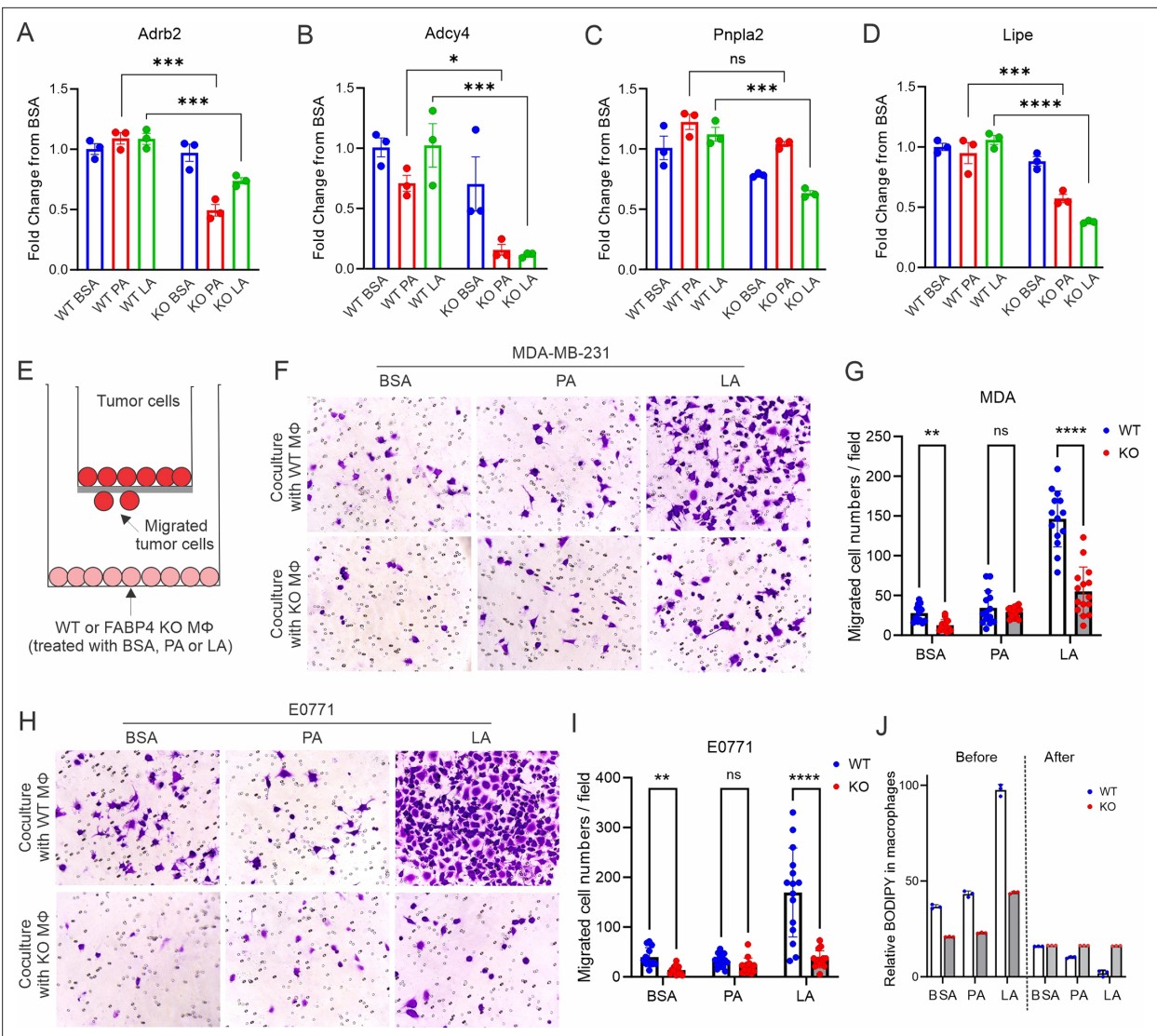

**Figure 6.** FAPB4 expression in macrophages promotes lipolysis and breast cancer cell migration. (**A–D**) Realtime PCR analysis of expression of Adrb2 (**A**), Adcy4 (**B**), Pnpla2 (**C**) and Lipe (**D**) in FABP4 WT and KO macrophages treated with BSA, PA, or LA (400 μM). (**E**) Transwell measurement of migration of breast cancer cells cocultured with FA- or BSA-treated FABP4 WT or KO macrophages (Mφ). (**F–I**) FABP4 WT or KO macrophages were treated with 100 μM BSA, PA, or LA for 4 hr. Fatty acids in the culture medium were washed away with FBS-free RPMI-1640. Breast cancer cells were added to a transwell and cocultured with these different FA-or BSA-treated FABP4 WT or KO macrophages for 24 hr. The migrated tumor cells were stained and quantified. Migrated MDA-MB-231 cells were shown in panels F and G. Migrated E0771 cells were shown in panels H and I. (**J**) FABP4 WT and KO macrophages were treated with indicated FAs or BSA for 4 hr. Flow cytometric staining of BODIPY levels in WT and KO macrophages before and after coculture with E0771 tumor cells for 24 hr. Data are shown as mean ± SD in panels A-D, G, I, J (* p≤0.01, p** p≤0.01, *** p≤0.001, ns, non-significant as compared to the control group or FABP4 KO group, unpaired Student t test).

The online version of this article includes the following figure supplement(s) for figure 6:

**Figure supplement 1.** FABP4 promotes lipolysis and tumor migration.

dietaryguidelines.gov/). However, data from the present study raise the concerns about the consumption of unsaturated fats, especially LA, in patients with breast cancer. Unsaturated FAs can induce lipid accumulation in macrophages, which can subsequently be mobilized to facilitate breast cancer migration and metastasis. Furthermore, we demonstrated that FABP4 plays a critical role in facilitating unsaturated FA-induced storage and lipolysis. Upon activation of LA, FABP4 traffics between the cytosol and nucleus to mediate lipid accumulation. In the presence of external factors (*e.g.*, tumors), FABP4 can be secreted from macrophages during lipolysis, which in turn enhances the utilization of

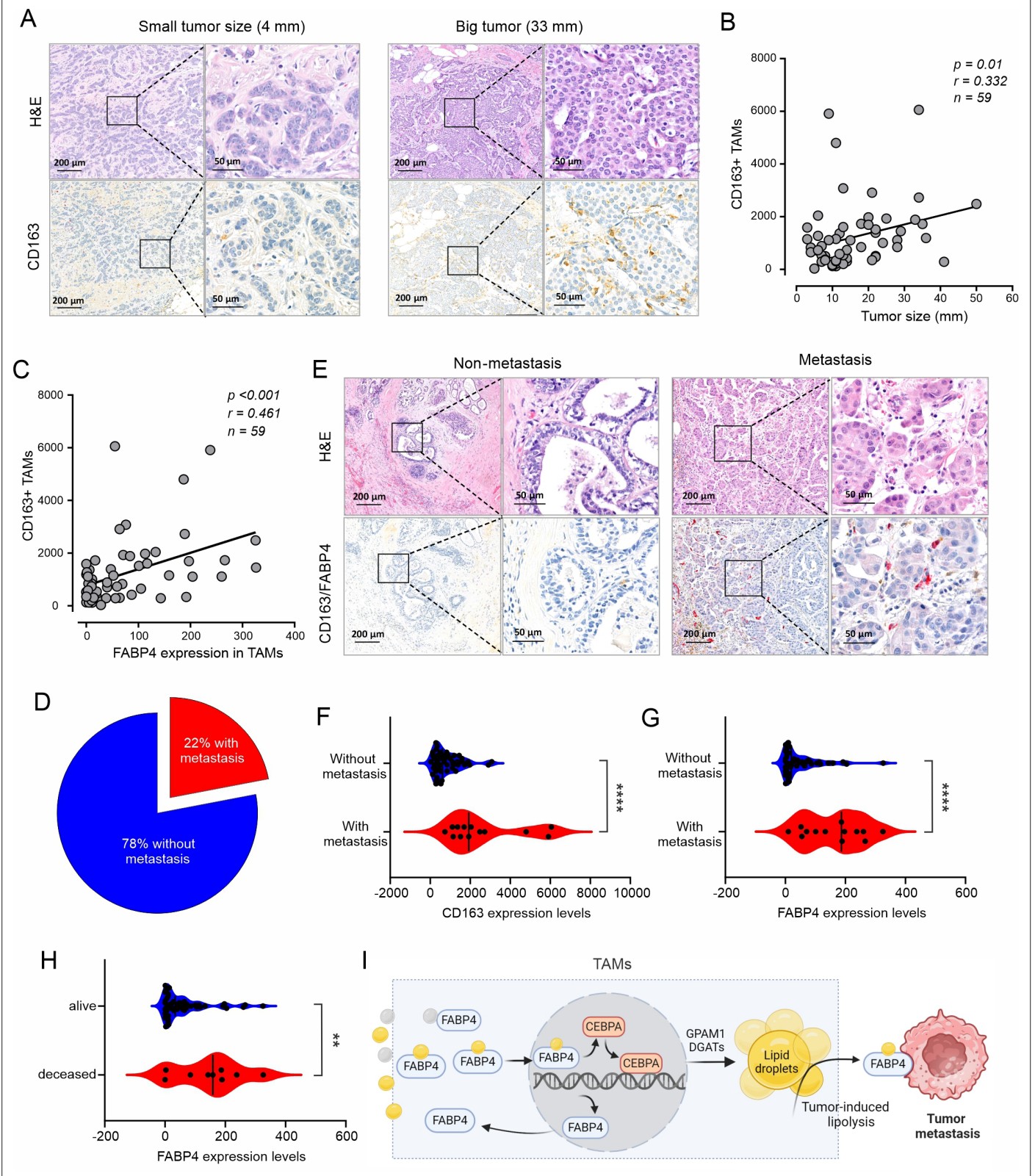

**Figure 7.** High expression of FABP4 in TAMs is associated with more metastasis of breast cancer. (**A**) Comparison of H&E and CD163 staining (brown) between an example of small and large breast cancer tumors in breast cancer patients. (**B**) Spearman correlation analysis between breast cancer tumor size and CD163 +TAM staining. (**C**) Expression of FABP4 and CD163 was highly correlated as analyzed by the Spearman correlation analysis in breast cancer tissues. (**D**) Pie chart showing the percentage of breast cancer patients with or without metastasis. (**E**) Analysis of the staining of H&E, CD163

*Figure 7 continued on next page*

*Figure 7 continued*

(brown), FABP4 (red) in primary breast tumors of patients with and without metastasis. (**F**) Analysis of CD163 expression levels between primary breast tumors of patients with and without metastases. (**G**) Analysis of FABP4 expression levels between primary breast tumors of patients with and without metastases. (**H**) Analysis of FABP4 expression levels between alive and deceased breast cancer patients. (**I**) Scheme of how FABP4 mediates unsaturated FA (yellow)-induced lipid storage and lipolysis in TAMs. When TAMs are exposed to dietary saturated (gray) or unsaturated (yellow) FAs, unsaturated FAs, but not saturated ones, induce FABP4 nuclear translocation and upregulate FABP4 and CEBPA-mediated transactivation of GPAM1 and DGATs, promoting lipid storage as lipid droplets. Once tumor-induced lipolysis occurs, FABP4/unsaturated FAs are secreted from TAMs to induce tumor migration and metastasis. Data are shown as mean ± SD in panel F-H (p** p0.01, *** p≤0.001, *** p≤0.001, unpaired Student t test).

The online version of this article includes the following figure supplement(s) for figure 7:

**Figure supplement 1.** Association of TAMs with obesity and survival of breast cancer patients.

FA ligands by cancer cells. Thus, blocking FABP4 lipid binding and secretion might represent novel strategies for preventing lipid utilization by cancer cells.

In summary, our studies have several implications. First, as professional phagocytes, macrophages uptake different lipids. Unlike saturated FAs, unsaturated FAs are preferentially accumulated in macrophages through the FABP4/CEBPα pathway. Secondly, TAMs with lipid accumulation facilitate breast cancer progression through the FABP4-dependent lipolysis and lipid utilization pathways. Thirdly, secreted FABP4/FAs might offer a previously underappreciated mechanism for promoting breast tumor metastasis potential (*Figure 7I*). Thus, developing therapies that block the activity of the secreted FABP4 might provide a new avenue for the treatment of breast cancer.

## Limitations of the study

In this study, we have identified FABP4 as a crucial lipid chaperone responsible for mediating lipid accumulation and lipolysis in TAMs. However, the mechanisms by which FABP4/FA complexes secreted from TAMs interact with tumor cells and mediate tumor metastasis remain unknown. While some potential FABP4 receptors, such as membrane phospholipids (*Hao et al., 2018b*), desmoglein 2 (*Chen et al., 2023*), cytokeratin 1 (*Saavedra et al., 2015*), have been reported in various cell types, further investigations are needed to determine how FABP4 and its lipid ligands facilitate tumor metastasis - either by initiating oncogenic signals, by providing energy sources or both.

## Materials and methods

### Mice

*Fabp4⁻/⁻* mice and their wild-type (WT) mice (C57B/6 background) were maintained and bred under specific pathogen free conditions with water and regular chow ad libitum in the animal facility of the University of Iowa in accordance with the approved protocol by the University of Iowa Institutional Animal Care and Use Committee (#1042385). As breast cancer mainly occurs in female, female mice at 8–12 weeks age were used for collection of primary peritoneal macrophages, splenic macrophages and culture of BMMs.

### Patient specimens

Fresh or formalin-fixed, paraffin-embedded breast cancer tissues were collected from the Breast and Molecular Epidemiology Resources (BMER) or Department of Pathology, University of Iowa. Breast sections were cut and transported to the study team without further patient identifiers. The University of Iowa Institutional Review Board approved the study protocol (#201003791, 202107133) prior to sample acquisition.

### Cell culture

WT and age matched *Fabp4⁻/⁻* mice were euthanized by CO2 inhalation and death was confirmed via subsequent cardiac puncture. Peritoneal macrophages were isolated by injecting 10 mL PBS (Gibco) into the mouse peritoneal cavity and incubated with 3 min with periodic abdominal palpation. EtOH was used to sterilize the surgical site and the skin of the abdomen was reflected. Using a 10 mL syringe coupled with a 25 G needle (Becton Dickinson and Company), PBS from the peritoneal cavity was aspirated, moved to a 15 mL conical (Avantor) tube, centrifuged, and resuspended in RPMI media with 10% FBS. To obtain bBMMs, femurs were dissected, muscle tissue was thoroughly removed, and

marrow was flushed with PBS into a 15 mL conical tube by way of a 10 mL syringe coupled with a 25 G needle. Isolated marrow cells were resuspended in red blood cell lysis buffer (Tonbo Biosciences) for 15 min on ice, washed with PBS, filtered through 40 µm cell strainers (VWR) and plated on 100 mm culture dishes (Greiner Bio-One) at a concentration of $5 \times 10^6$ cells/mL in RPMI media with 10% FBS and 30 ng/mL recombinant mouse macrophage colony-stimulating factor (M-CSF; BioLegend). After 48 hr and again after 120 hr from initial plating cell culture media was replaced with fresh media containing M-CSF. Cells were scrutinized after 168 hr in culture.

Immortalized macrophage cell lines were established from FABP4$^{-/-}$ (FABP4$^{-/-}$ macrophages) or WT mice (WT macrophages) as described previously (*Zhang et al., 2017c*; *Makowski et al., 2005*; *Clemons-Miller et al., 2000*). Briefly, BMMs were culture as outlined above however following differentiation J2-virus conditioned media was used for 2 hr and replaced with M-CSF media. This procedure was repeated after 24 hr. Subsequently M-CSF media was removed step-wise over the course of 2 weeks at which point cells could be effectively passages. This procedure was completed for marrow cells obtained from a C57Bl/6 J mouse as well as aged match FABP4$^{-/-}$ mouse. Breast cancer cell migration assays were completed using E0771 mouse mammary gland cells and MDA-MB 231 human mammary gland cells (ATCC).

## Free fatty acid (FFA) preparation

Due to their insolubility, all FFAs were conjugated with bovine serum albumin (BSA) as described previously (*Zhang et al., 2017c*; *Zhang et al., 2014*; *Jin et al., 2021b*; *Zhang et al., 2017a*). Briefly, all FFAs used in the current study were purchased from Nu-Chek Prep (MN). PA (5 mM, S1109), SA (5 Mm, S-1111), OA (5 Mm, S-1120) or LA (5 Mm, S-1127) were respectively conjugated with 2 mM of endotoxin-free, FA-free BSA (Proliant, Cat# 69760) in PBS. FA-BSA conjugates were sonicated until dissolved. All FA-BSA conjugates or BSA control were filtered through a 0.2 µM sterile filter before use in cellular studies.

## RNA*i*

Predesigned duplex small interfering RNA was obtained from Integrated DNA Technologies (IDT) specific to C/EBPα and C/EBPβ. J2-Immortalized macrophages were transfected using jetPRIME (Polyplus) as instructed by the manufacture. Efficacy of RNAi was confirmed by RT-PCR for each experiment.

## Quantitative RT-PCR

Cells were lysed and RNA was extracted using the PureLink RNA mini kit (Invitrogen) as per manufactures instructions. RNA concentration was determined and standardized by the aid of a photospectrometer and complementary DNA was synthesized by use of QuantiTect reverse transcription kit (QIAGEN). Power SYBR Green PCR Master Mix (Applied Biosystems) was included with target primers (*Supplementary file 3*) on MicroAmp optical 384-well reaction plates (Applied Biosystems) and a QuantStudio 7 Flex (Thermo Fisher) and the provided softeware (Design and Analysis Software V2.3, Thermo Fisher) was used to determine relative mRNA concentration by way of ΔΔCt method with hprt1 as an internal control. Standard PCR reaction parameters were used: 95 °C hold stage for 10 min, 40 cycles of PCR reaction of 95 °C for 15 s and 60 °C for 1 min, and a melt curve stage used 0.5 °C steps from 60°C to 65°C.

## Seahorse cell mito stress

The Seahorse mitochondrial stress test (Agilent) was performed as specified by the manufacturer. Briefly, cells were plated at a concentration of $5 \times 10^4$ per well in RPMI media with 10% FBS. Before the assay cells were wash twice with RPMI-XF media with no FBS and incubated at 37 °C for 15 min during equipment initialization. supplemented with 100 µM BSA, PA, SA, OA, or LA. The acute injection protocol was utilized where some wells (n=5 for each FA for each genotype) received injections of RPMI-XF supplemented with 200 µM BSA, PA, SA, OA, or LA, and all wells subsequently received injections of 2.5 µM oligomycin, 1 µM FCCP, and finally 1 µM Rotenone/Antimycin/Hoechst. Assays were ran using the XFe96 extracellular flux analyzer (Agilent) coupled with automated normalization based on live cell number at the conclusion of the assay.

## Breast cancer tissue immunohistochemistry

Formalin fixed Paraffin Embedded (FFPE) tissue specimens were deparaffinized in xylene and rehydrated through exposure to graded ethanol solutions. Antigen retrieval was performed by incubating the slides in 10 mM Citrate buffer pH6.0 at 95 °C for 15 min. Then, quenching of endogenous peroxidase activity was performed by incubating the slides in BLOXALL Endogenous Blocking Solution (Vector Laboratories, SP-6000) for 10 min at room temperature. Slides were incubated with a primary antibody overnight at 4 °C, a secondary antibody for 30 min at room temperature (All the primary and secondary antibodies used in this study are summarized in key resources table). DAB peroxidase substrate (Vector Laboratories, SK-4100) or Vector Red alkaline phosphatase substrate (Vector Laboratories, SK-5100) were used for staining. Counterstain was performed by incubating the slides in Hematoxylin (Leica Gill III 3801541) for 10 s at room temperature. Slides were mounted with VectaMount Express Mounting Medium (vector laboratories, H-5700–60), and were scanned by slide scanner (Leica Aperio GT 450) for quantification analysis. Hot spot quantification of CD163 and FABP4 expression in macrophages (per 1x2 mm$^2$ area within the tumor tissue) was automatically calculated by the HALO software (Indica Labs).

## Transmission electron microscopy

Cells were fixed with 2.5% glutaraldehyde (in 0.1 M Sodium cacodylate buffer [pH 7.4]) overnight at 4 °C. samples were postfixed with 1% Osmium tetroxide for 1 hr and then rinsed in 0.1 M Sodium cacodylate buffer. Following serial alcohol dehydration (50%, 75%, 95%, and 100%), the samples were embedded in Epon 12 (Ted Pella, Redding, CA). Ultramicrotomy was performed, and ultrathin sections (70 nm) were poststained with uranyl acetate and lead citrate. Samples were examined with a Hitachi HT-7800 transmission electron microscope (TEM) (Tokyo, Japan).

## Immunogold analysis for FABP4 in macrophages

Cells were fixed with 4% paraformaldehyde (PFA), and then were dehydrated in graded concentrations of ethanol and embedded in LR White resin (Electron Microscopy Sciences, 14380). Ultrathin sections (70 nm) on grids were incubated with blocking solution (Aurion, 905.003) for 1 hr, and then incubated with primary antibody overnight at 4 °C, and then incubated with secondary antibody (Aurion, Rabbit-anti-Goat IgG (H&L) Ultra Small) for 2 hr. Grids were re-fixed with 1% glutaraldehyde for 15 min and were incubated in sliver enhancement (Aurion, 500.033) for 25 min. Sections were then stained with uranyl acetate and lead citrate and examined with a Hitachi HT-7800 transmission electron microscope (TEM; Tokyo, Japan).

## Oil Red O staining

For each use Oil Red O powder (Alfa Aesar) was solubilized in 100% isopropanol (75 mg in 25 mL) and mixed on a orbital shaker for 30 min to make a stock solution. For a working solution Oil Red O stock solution was used at a ratio of 3 parts Oil Red O to 2 parts ddH$_2$O and mixed on an orbital shaker for 10 min. The working solution was filtered twice through 0.2 μm syringe filter and used immediately on 4% PFA fixed cells. During the second of three washes with ddH$_2$O, DAPI was included at 1:5000 for nuclear identification. Images were taken with an ECHO Revolution2 (ECHO) epifluorescence microscope and processed using ImageJ/Fiji. Following image collection, for the purpose of semi-quantitative analysis, samples were washed with 60% isopropanol three times for 5 min each and the residual Oil Red O was solubilized with 100% Isopropanol and absorbance was assessed by spectrophotometer at 492 nm, 100% isopropanol was used as a background.

## Flow cytometry

The following anti-mouse conjugated antibodies were used to label macrophages surface receptors from peritoneal and bone marrow derived cells: PE/Cyanine7-F4/80, BV711-CD11c, BV605-I-A/I-E (BioLegend), BUV737-CD11b, and BUV563-CD45 (BD Biosciences). Unconjugated antibodies were used for quantification of intracellular markers: m/rFABP4 (R&D Systems), GPAM (Invitrogen), DGAT1 (Sigma-Aldrich), C/EBPα (Cell Signaling), C/EBPβ (DSHB). The following organelle dyes were used: BODIPY 493/503 (Lipid Droplets; Molecular Probes), Cell Navigator Live Cell Endoplasmic Reticulum Staining Kit Red (ATT Bioquest), and LysoTracker Blue DND-22 (Invitrogen). Dead cells were identified and removed from analysis via Ghost Dye Violet 510 (Tonbo Biosciences). Briefly, live cells were

scrutinized unless fixation was required for intracellular targets in which case 4% PFA was used for 30 min to fix the cells and 0.2% Triton X-100 in PBS was used for permeabilization. Antibodies or organelle dyes were incubated with cells for 30 min on ice in the dark. When necessary secondary antibodies were incubated with cells for 1 hr on ice in the dark. Data was acquired using a Cytek Aurora 5-laser flow cytometer or ImageStream cytometer (CytekBio) and subsequently analyzed FlowJo V10 (BD Biosciences).

## Migration assay

For the migration assay, we treated WT and FABP4 KO macrophages with BSA, PA, or LA (100 µM) for 4 hr in a 24-well plate. After washing them three times with PBS, we seeded E0771 or MDA-MB 231 cells (5×104) into transwell inserts with a pore size of 8 µm (Corning). These cells were incubated with the FA-treated macrophages, respectively, to allow tumor cells to migrate. After a 24 hr culture at 37 °C, we removed the tumor cells inside the transwells that had not migrated using cotton swabs. The tumor cells that had migrated to the outside of the insert membrane were fixed with 4% PFA for 20 min and then stained with 0.1% crystal violet for 10 min. We counted the numbers of migrated E0771 and MDA-MB 231 cells in 12 random high fields under a microscope. Each assay was performed in triplicate wells. Additionally, in some experiments, we used conditional media (RPMI-1640 without FBS) collected from BSA or FA-treated WT and KO macrophages to coculture with tumor cells and observe medium-induced tumor cell migration.

## Single-cell RNA-seq assay

Macrophages from mouse spleen were used for the 10 x Genomics 3' expression single cell assay. Briefly, splenocytes were isolated from two pairs of age-matched WT and FABP4-/- female mice. Dead cells were excluded using Zombie-Violet and anti-CD16/32 antibody was used to block Fc receptors before additional staining with Alexa Fluor 488-conjugated anti-mouse F4/480 antibody and APC-conjugated anti-mouse CD11b antibody. The F4/80+/CD11b+/Zombie-Voilet- cells were sorted using a FACS Aira III instrument and resuspended at a concentration of 1000 cell/µL cold PBS with 0.04% non-acetylated BSA. An equal number of 5000 targeted cells from each sample were prepared to create GEMs. The GEM generation/barcoding, post GEM-reverse transcription cDNA amplification, and library construction were performed according to the manufacturer's guidelines.

## Single-cell data analysis

Single-cell gene counting was performed by Cell Ranger (10 X Genomics, version 7.1.0) using the refdata-gex-mm10-2020-A reference transcriptome. The resulting data matrices were subsequently imported into R (version 4.2.3) and analyzed using the Seurat package (version 4.9.9). Cells with less than 200 features or with percent mitochondrial gene expression greater than 5% were excluded from the analysis. The data across samples was integrated using the Integrate Data function. Gene expression was normalized and scaled using the default parameters. Based on visual inspection of the elbow plot, the first 30 Principal Components (PC) were used in UMAP-based dimensional reduction. The FindClusters function, with a resolution of 0.5, was then used to assign cells to clusters. The FindMarkers function was used to identify genes differentially expressed between wild-type and knockout samples or between different clusters within the same sample type. Feature plots and violin plots were used to visualize the results.

## ELISA for FABP4 analysis

FABP4 excreted from cultured cells were analyzed with CircuLex Mouse FABP4 ELISA Kit (Medical and Biological Laboratories) as instructed by the manufactures. Briefly, for the ELISA 100 µL of conditioned media was incubated in a 96-well plate precoated with FABP4 antibody, washed four times, incubated with the supplied secondary antibody, washed, and incubated with the developer to finally analyze with a spectrophotometer using a ratio of the absorbance at 450 nm and 550 nm.

## Quantification and statistical analysis

All data were presented as the mean ± SD unless notified specifically. All experiment as indicated were performed by at least three independent experiments or technical replicates. For in vitro experiments, a two-tailed, unpaired student t-test, two-way ANOVA followed by Bonferroni's multiple

comparison test, were performed by GraphPad Prism 9. For in vivo experiments analyzing associations of FABP4, TAMs and other factors in breast cancer patients, the Kruskal-Wallis test was applied to compare whether the outcome variable is significantly different across the different levels of a categorical predictor variable. Multiple linear regression models were employed to examine the association between the predictor variables and the outcome, where log transformation and Box Cox transformation were used to meet the underlying model assumption for normality. All statistical analyses were carried out in the statistical analysis software R. A statistics test is claimed to be significant if the p-value is less than 0.05.

## Acknowledgements

We thank the Iowa Institute of Human Genetics at the University of Iowa for single cell RNA-sequencing and analysis. We also thank the Holden Comprehensive Cancer Center's P30 grant which partially funds the Breast Molecular Epidemiology Resource (BMER) (P30CA086862). BL thank the funding support from NIH grants U01CA272424, R01CA180986 and R01AI137324. This material should not be interpreted as representing the viewpoint of the U.S. Department of Health and Human Services, the National Institutes of Health, or the National Cancer Institute.

## Additional information

### Funding

| Funder | Grant reference number | Author |
| --- | --- | --- |
| National Institutes of Health | U01CA272424 | Bing Li |
| National Institutes of Health | R01CA180986 | Bing Li |
| National Institutes of Health | R01AI137324 | Bing Li |

The funders had no role in study design, data collection and interpretation, or the decision to submit the work for publication.

### Author contributions

Matthew Yorek, Data curation, Investigation, Methodology; Xingshan Jiang, Shanshan Liu, Data curation, Formal analysis, Investigation, Methodology; Jiaqing Hao, Jianqiang Shao, Data curation, Methodology; Jianyu Yu, Data curation; Anthony Avellino, Methodology; Zhanxu Liu, Henry Keen, Formal analysis, Methodology; Melissa Curry, Anand Kanagasabapathy, Yiqin Xiong, Resources, Writing – review and editing; Maying Kong, Formal analysis, Methodology, Writing – review and editing; Edward R Sauter, Conceptualization, Writing – review and editing; Sonia L Sugg, Conceptualization, Funding acquisition, Writing – review and editing; Bing Li, Conceptualization, Supervision, Funding acquisition, Investigation, Writing – original draft

### Author ORCIDs

Bing Li https://orcid.org/0000-0001-5363-7070

### Ethics

All of the animals were handled according to approved institutional animal care and use committee (IACUC) protocols (#1042385).

Reviewer #1 (Public review): https://doi.org/10.7554/eLife.101221.2.sa1
Reviewer #2 (Public review): https://doi.org/10.7554/eLife.101221.2.sa2
Reviewer #3 (Public review): https://doi.org/10.7554/eLife.101221.2.sa3
Author response https://doi.org/10.7554/eLife.101221.2.sa4

## Additional files

### Supplementary files

• Supplementary file 1. Patient information.

• Supplementary file 2. Summary statistics for categorical variables and their association with outcome variables.

• Supplementary file 3. Realtime PCR primer sequences.

• MDAR checklist

### Data availability

All data generated or analyzed during this study are included in the manuscript and supporting files.

The following previously published datasets were used:

| Author(s) | Year | Dataset title | Dataset URL | Database and Identifier |
|---|---|---|---|---|
| Clarke C, Madden SF, Doolan P, Joyce H, Aherne S, O' Driscoll L, Gallagher WM, Hennesy B, Moriarty M, Crown J, Kennedy S, Clynes M | 2013 | Breast Cancer Gene Expression Analysis | https://www.ncbi.nlm.nih.gov/geo/query/acc.cgi?acc=GSE42568 | NCBI Gene Expression Omnibus, GSE42568 |
| Schaefer C, Kemmner W | 2011 | Expression profiling of human DCIS and invasive ductal breast carcinoma | https://www.ncbi.nlm.nih.gov/geo/query/acc.cgi?acc=GSE21422 | NCBI Gene Expression Omnibus, GSE21422 |
| Cuadros M, Cano C, Lopez FJ, Blanco A, Concha A | 2011 | Identifying breast cancer biomarkers | https://www.ncbi.nlm.nih.gov/geo/query/acc.cgi?acc=GSE29431 | NCBI Gene Expression Omnibus, GSE29431 |
| Schroeder M, Haibe-Kains B, Culhane A, Sotiriou C, Bontempi C, Quackenbush J | 2002 | Genexpression dataset published by van't Veer et al. [2002] and van de Vijver et al. [2002] (NKI) | https://doi.org/10.18129/B9.bioc.breastCancerNKI | BreastCancerNKI, 10.18129/B9.bioc.breastCancerNKI |
| Weigman VJ, Perou CM | 2011 | Basal-like Breast Cancer DNA copy number losses identify genes involved in genomic instability, response to therapy, and patient survival | https://www.ncbi.nlm.nih.gov/geo/query/acc.cgi?acc=GSE10893 | NCBI Gene Expression Omnibus, GSE10893 |
| Prat A, Perou CM | 2010 | Phenotypic and Molecular Characterization of the Claudin-low Intrinsic Subtype of Breast Cancer | https://www.ncbi.nlm.nih.gov/geo/query/acc.cgi?acc=GSE18229 | NCBI Gene Expression Omnibus, GSE18229 |
| Chanrion M, Vincent N, Fontaine H, Salvetat N, Bibeau F, MacGrogan G, Mauriac L, Katsaros D, Molina F, Theillet C, Darbon JM | 2008 | A gene expression signature predicting the recurrence of tamoxifen-treated primary breast cancer | https://www.ncbi.nlm.nih.gov/geo/query/acc.cgi?acc=GSE9893 | NCBI Gene Expression Omnibus, GSE9893 |

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

# Appendix 1

## Appendix 1—key resources table

| Reagent type (species) or resource | Designation | Source or reference | Identifiers | Additional information |
|---|---|---|---|---|
| Antibody | PE/Cyanine7 rat anti-mouse F4/80 monoclonal | BioLegend | Cat#123114 | 1:500 |
| Antibody | Brilliant Violet 711 Armenian Hamster anti-mouse CD11c monoclonal | BioLegend | Cat#117349 | 1:500 |
| Antibody | Brilliant Violet 605 rat anti-mouse I-A/I-E monoclonal | BioLegend | Cat#107639 | 1:500 |
| Antibody | BUV737 rat anti-mouse CD11b monoclonal | BD Biosciences | Cat#612800 | 1:500 |
| Antibody | BUV563 rat anti-mouse CD45 monoclonal | BD Biosciences | Cat#752412 | 1:500 |
| Antibody | Goat anti-human FABP4polyclonal | R&D Systems | Cat#AF3150 | 1:300 |
| Antibody | Goat anti-mouseFABP4 polyclonal | R&D Systems | Cat#AF1443 | 1:300 |
| Antibody | Rabbit anti-human/mouse GPAM polyclonal | Invitrogen | Cat#PA5-20524 | 1:300 |
| Antibody | Goat anti-human/mouse DGAT1 polyclonal | Sigma-Aldrich | Cat#SAB2500307 | 1:300 |
| Antibody | Rabbit anti-human/mouse C/EBPα polyclonal | Cell Signaling Technology | Cat#2295 | 1:300 |
| Antibody | Mouse anti-human C/EBPβ monoclonal | DSHB | Cat#PCRP-CEBPB-3D10 | 1:300 |
| Antibody | Rabbit anti-CD163 human/mouse monoclonal | Abcam | Cat#EPR19518 | 1:300 |
| Antibody | Alexa Fluor 594-conjugated AffiniPure mouse anti-goat IgG polyclonal | Jackson ImmunoResearch | Cat#205-605-108 | 1:2000 |
| Antibody | Alexa Fluor 647-conjugated AffiniPure donkey anti-goat IgG polyclonal | Jackson ImmunoResearch | Cat#705-586-147 | 1:2000 |
| Antibody | Alexa Fluor Plus 488 donkey anti-goat IgG polyclonal | Invitrogen | Cat#A32814 | 1:2000 |
| Antibody | Alexa Fluor 647-conjugated AffiniPure Goat anti-rabbit IgG polyclonal | Jackson ImmunoResearch | Cat#111-605-003 | 1:2000 |
| Antibody | Alexa Fluor 546 goat anti-rabbit IgG polyclonal | Invitrogen | Cat#A11035 | 1:2000 |
| Antibody | Alexa Fluor 488 goat anti-mouse IgG polyclonal | Invitrogen | Cat#A11029 | 1:2000 |
| Antibody | Alexa Fluor 546 goat anti-mouse IgG polyclonal | Invitrogen | Cat#A11030 | 1:2000 |
| Chemical compound, drug | Ghost Dye Violet 510 Viability Dye | Tonbo Biosciences | Cat#13–0870 T100 | 1:200 |
| Chemical compound, drug | Bodipy 493/503 | Molecular Probes | Cat#D3922 | Final concentration: 5 µM |
| Chemical compound, drug | LysoTracker Blue DND-22 | Invitrogen | Cat#L7525 | Final concentration: 50 nM |
| Chemical compound, drug | Cell Navigator Live Cell Endoplasmic Reticulum Staining Kit Red | ATT Bioquest | Cat#22636 | 500 × |
| Chemical compound, drug | MitoSpy NIR DilC1(5) | BioLegend | Cat#424807 | Final concentration: 10 nM |
| Chemical compound, drug | HCS LipidTOX Green Neutral Lipid Stain | Invitrogen | Cat#H34475 | 1000 × |
| Chemical compound, drug | Hoechst 33342 Solution | Invitrogen | Cat#62249 | Final concentration: 0.1 µg/ml |
| Chemical compound, drug | DAPI solution (1 mg/mL) | Invitrogen | Cat#62248 | 1:1000 |
| Chemical compound, drug | Oil Red O powder | Alfa Aesar | Cat#A12989.14 | Final concentration: 1.8 mg/ml |
| Chemical compound, drug | Phosphate Buffered Saline pH 7.4 (PBS) | Gibco | Cat#10010–023 | |
| Chemical compound, drug | Paraformaldehyde Solution 4% in PBS | Thermo Fisher Scientific | Cat#J19943.K2 | |

*Appendix 1 Continued on next page*

*Appendix 1 Continued*

| Reagent type (species) or resource | Designation | Source or reference | Identifiers | Additional information |
| --- | --- | --- | --- | --- |
| Chemical compound, drug | Neutral Buffered Formalin Solution 10% in PBS | VWR | Cat#10790–714 | |
| Chemical compound, drug | Triton X-100 | Sigma-Aldrich | Cat#X100-500ML | Final concentration: 0.2% |
| Chemical compound, drug | ProLong Diamond Antifade Mountant | Invitrogen | Cat#P36961 | |
| Chemical compound, drug | RPMI Medium 1640 | Gibco | Cat#11875–093 | |
| Chemical compound, drug | Seahorse XF RPMI Meduim pH7.4 | Agilent Technologies | Cat#103576–100 | |
| Chemical compound, drug | Seahorse XF calibrant solution | Agilent Technologies | Cat#100840–000 | |
| Chemical compound, drug | Penicillin-Streptomycin | Thermo Fisher Scientific | Cat#15140122 | 100 × |
| Chemical compound, drug | Fetal Bovine Serum | R&D Systems | Cat#S11550 | |
| Chemical compound, drug | Goat Serum | Thermo Fisher Scientific | Cat#31873 | |
| Chemical compound, drug | Palmitate | Nu-Chek Prep, Inc | S-1109 | 5 mM in stock |
| Chemical compound, drug | Stearate | Nu-Chek Prep, Inc | S-1111 | 5 mM in stock |
| Chemical compound, drug | Oleate | Nu-Chek Prep, Inc | S-1120 | 5 mM in stock |
| Chemical compound, drug | Linoleate | Nu-Chek Prep, Inc | S-1127 | 5 mM in stock |
| Chemical compound, drug | Bovine Serum Albumin | Proliant | Cat# 69760 | 5 mM in stock |
| Chemical compound, drug | Poly-L-lysine | Sigma-Aldrich | Cat#P9155 | 0.5 ml of a 0.1 mg/ml solution to coat 25 cm$^2$ |
| Chemical compound, drug | jetPRIME | Sartorius | Cat#101000046 | |
| Chemical compound, drug | Power SYBR Green PCR Master Mix | Applied Biosystems | Cat#4368708 | |
| Chemical compound, drug | RBC Lysis Buffer | Tonbo Biosciences | Cat#TNB-4300-L100 | |
| Commercial assay, kit | Chromium NextGEM Chip G Single Cell Kit | 10 X Genomics | Cat#1000127 | |
| Commercial assay, kit | Dual Index Kit TT Set A | 10 X Genomics | Cat#1000215 | |
| Commercial assay, kit | Chromium NextGEM Single Cell 3'Kit v3.1 | 10 X Genomics | Cat#1000269 | |
| Commercial assay, kit | Magnetic Separator | 10 X Genomics | Cat#120250 | |
| Commercial assay, kit | Non-acetylated BSA | 10 X Genomics | Cat#B9000S | |
| Commercial assay, kit | Seahorse XF Cell Mito Stress Test Kit | Agilent Technologies | Cat#103015–100 | |
| Commercial assay, kit | Seahorse XF96 Cell Culture Plate | Agilent Technologies | Cat#101085–004 | |
| Commercial assay, kit | Seahorse XFe96 Extracellular flux assay kits | Agilent Technologies | Cat#102601–100 | |
| Commercial assay, kit | PureLink RNA mini kit | Invitrogen | Cat#12183025 | |
| Commercial assay, kit | QuantiTect reverse transcription kit | Qiagen | Cat#205314 | |
| Commercial assay, kit | Zombie Violet Fixable Viability Kit | Biolegend | Cat#423113 | 1:1000 |
| Commercial assay, kit | ImmPRESS HRP Horse Anti-Rabbit IgG Polymer Detection Kit, Peroxidase | Vector | Cat#MP-7401–50 | |
| Commercial assay, kit | BLOXALL Endogenous Blocking Solution, Peroxidase and Alkaline Phosphatase | Vector | SP-6000–100 | |
| Commercial assay, kit | CircuLex Mouse FABP4/A-FABP ELISA Kit | CircuLex/MBL | Cat#CY-8077 | |
| Sequence-based reagent | DsiRNA for mouse C/EBPα | IDT | Cat#mm.Ri.Cebpa.13.1 | Transfection at final concentration of 50 nM |
| Sequence-based reagent | DsiRNA for mouse C/EBPβ | IDT | Cat#mm.Ri.Cebpb.13.1 | Transfection at final concentration of 50 nM |
| Peptide, recombinant protein | Recombinant Mouse M-CSF | BioLegend | Cat#576406 | Final concentration: 30 ng/ml |

*Appendix 1 Continued*

| Reagent type (species) or resource | Designation | Source or reference | Identifiers | Additional information |
|---|---|---|---|---|
| Recombinant DNA reagent | Microarray data (Breast Cancer Gene Expression Analysis) | PMID:23740839 | GSE42568 | |
| Recombinant DNA reagent | Microarray data (Expression profiling of human DCIS and invasive ductal breast carcinoma) | PMID:21468687 | GSE21422 | |
| Recombinant DNA reagent | Microarray data (Identifying breast cancer biomarkers) | PMID:141503 | GSE29431 | |
| Recombinant DNA reagent | NKI | PROGeneV2 | https://bioconductor.org/packages/breastCancerNKI | |
| Recombinant DNA reagent | Microarray data | PROGeneV2 | GSE10893-GPL887 | |
| Recombinant DNA reagent | Microarray data | PROGeneV2 | GSE18229-GPL887 | |
| Recombinant DNA reagent | Microarray data | PROGeneV2 | GSE9893 | |
| Cell line (mouse) | Bone Marrow Derived Macrophage-J2 Immortalized | This Paper | N/A | See the Methods details – Cell lines |
| Cell line (mouse) | FABP4$^{-/-}$ Bone Marrow Derived Macrophage-J2 Immortalized | This Paper | N/A | See the Methods details – Cell lines |
| Cell line (mouse) | BMM | This paper | N/A | See the Methods details – Primary cells |
| Cell line (mouse) | Peritoneal Macrophage | This paper | N/A | See the Experimental model and study participant details – Mice |
| Cell line (mouse) | E0771 | ATCC | Cat#CRL-3461 | |
| Cell line (human) | MDA-MB 231 | ATCC | Cat#HTB-26 | |
| Strain, strain background | Mouse:C57Bl/6 J | Jackson Laboratory | JAX 000664 | |
| Strain, strain background | Mouse:FABP4$^{-/-}$ | This Paper | | See the Experimental model and study participant details – Mice |
| Sequence-based reagent | Real-time PCR Primers | **Supplementary file 3** | N/A | |
| Software and algorithms | FlowJo v10 | BD Biosciences | https://www.flowjo.com/ | |
| Software, algorithm | GraphPad Prism 8 | GraphPad Software | https://www.graphpad.com/ | |
| Software, algorithm | ImageJ (Fiji edition) | NIH | https://imagej.net/software/fiji/ | |
| Software, algorithm | Design and Analysis Software V2.3 | Thermo Fisher | http://www.thermofisher.com/us/en/home/global/forms/life-science/quantstudio-6-7-pro-software.html | |
| Software, algorithm | Seahorse Wave | Agilent Technologies | https://www.agilent.com/zh-cn/product/cell-analysis/real-time-cell-metabolic-analysis/xf-software/seahorse-wave-desktop-software-740897 | |
| Software, algorithm | Web-based Gene Set Analysis Toolkit | WebGestalt | https://www.webgestalt.org/ | |
| Software, algorithm | VolcaNoseR2 | VolcaNoseR2 | https://huygens.science.uva.nl/VolcaNoseR2/ | |
| Software, algorithm | g:Profiler | ELIXIR | https://biit.cs.ut.ee/gprofiler/gost | |
| Software, algorithm | CiiiDER | CiiiDER | The program and documentation are available from https://www.ciiider.org/ and the source code is available at https://gitlab.erc.monash.edu.au/ciiid/ciiider (**Gearing, 2020**). | Refer to https://journals.plos.org/plosone/article?id=10.1371/journal.pone.0215495 |
| Software, algorithm | CorelDRAW X7 | CorelDRAW | https://www.coreldraw.com/en/ | |
| Software, algorithm | Halo Software | Indica Labs | V3.6.4134.263 | |
| Other | Sterile 24 well cell culture plate | Corning | Cat#3526 | See the Method details – Migration assay |
| Other | Sterile 100 mm cell culture dish | Greiner Bio-One | Cat#664160 | See the Method details – Primary cells |
| Other | Sterile 25Gx1" Needle | Becton, Diskinson and Company | Cat#305125 | See the Method details – Primary cells |

*Appendix 1 Continued on next page*

*Appendix 1 Continued*

| Reagent type (species) or resource | Designation | Source or reference | Identifiers | Additional information |
|---|---|---|---|---|
| Other | Sterile 10 mL syringe | Becton, Diskinson and Company | Cat#302995 | See the Method details – Primary cells |
| Other | Sterile 15 mL conical-bottom Centrifuge Tube | Avantor | Cat#525–1068 | See the Method details – Primary cells |
| Other | 40 µm cell strainer | VWR | Cat#BX15-1040 | See the Method details – Primary cells |
| Other | Transwell inserts 8.0 µm pore size | Falcon | Cat#353097 | See the Method details – Migration assay |
| Other | 5 mL Polystyrene round-bottom tube | Falcon | Cat#352058 | See the Method details – Flow cytometry |
| Other | MicroAmp Optical 384-well Reaction Plate | Applied Biosystems | Cat#4309849 | See the Method details – Quantitative RT-PCR |
| Other | QuantStudio 7 Flex Real-Time PCR system | Applied Biosystems | Cat#4485701 | See the Method details – Quantitative RT-PCR |
| Other | Seahorse XFe96 extracellular flux analyzer | Seahorse Biosciences | | See the Method details – Seahorse cell mito stress |
| Other | Cytek Aurora | CytekBio | | See the Method details – Flow cytometry |
| Other | Cytek Amnis ImageStream MkII Imaging Flow Cytometer | CytekBio | | See *Figure 1F* |
| Other | Synergy LX Multi-Mode Microplate Reader | BioTek | Cat#SLXFATS-SN | See the Method details – ELISA for FABP4 analysis |
| Other | Echo Revolution 2 microscope | ECHO | | See *Figure 6F* and 6 H |
| Other | Zeiss LSM880-airyscan | Zeiss | | See *Figure 2I*, *Figure 3B and L*, *Figure 4K and L*, and *Figure 5G* |
| Other | Electron Microscope | Hitachi | HT7800 | See the Method details – Transmission electron microscopy |
| Other | Slide Scanner | Leica | Aperio GT 450 | See *Figure 7* |

