## [Editor Report · eLife Assessment]

This **important** paper uses elegant models, including genetic knock outs, to demonstrate that FABP4 contributes to lipid accumulation in tumor-associated macrophages, which seems to increase breast cancer migration. While the work is of high interest, the strength of the evidence relating to some of the conclusions is **incomplete** and the paper would benefit from some refinement. The work will be of interest to those researchers trying to link metabolism, the immune system, and cancer.

---

## [Referee Report · Reviewer #1 (Public review)]

Summary:

In the manuscript "FABP4-mediated lipid accumulation and lipolysis in tumor-associated macrophages promote breast cancer metastasis", Yorek, et al. provide a novel mechanism explaining how unsaturated fatty acids induce macrophages to accumulate lipid droplets, which when contained in tumor-associated macrophages (TAMs) are associated with increased metastasis in breast cancer. The authors conclude that unsaturated fatty acids are transported into macrophages by the chaperone FABP4 where they induce C/EBPalpha expression and transcriptional activity resulting in upregulation of enzymes involved in triacylglycerol and lipid droplet biosynthesis. The resulting accumulation of lipid droplets in macrophages creates a store of fatty acids that can subsequently be released through FABP4-dependent lipolysis and thereby stimulate the migration of nearby breast cancer cells. While generally well-written and developed, there are a few concerns about the rigor of experimental evidence that supports some conclusions, including the existence of a FABP4-C/EBPalpha pathway. Overall, the mechanism identified is a valuable contribution to our understanding of how tumor-associated macrophages may influenced by available metabolites to promote the aggressiveness of certain cancers. FABP4 has the potential to be used as a novel biomarker of macrophage-induced cancer aggressiveness and/or a therapeutic target to prevent metastasis.

Strengths:

(1) The study is logically organized and provides extensive evidence in support of the overall model proposed.

(2) Multiple complementary techniques are used to identify and quantify lipid droplets.

(3) Primary macrophages and macrophage cell lines are used and provide consistent data.

(4) Knock-down and knock-out cells are used to assess the contributions of FABP4 and C/EBPalpha to gene expression.

(5) Public gene expression data (GEO, TCGA) is used effectively throughout.

Weaknesses:

(1) After Figure 1, a single saturated (palmitic acid; PA) and a single unsaturated (linoleic acid; LA) fatty acid are used for the remaining studies, bringing into question whether effects are in fact the result of a difference in saturation vs. other potential differences.

(2) While primary macrophages are used in several mechanistic studies, tumor-associated macrophages (TAMs) are not used. Rather, correlative evidence is provided to connect mechanistic studies in macrophage cell lines and primary macrophages to TAMs.

(3) C/EBPalpha and FABP4 clearly regulate LA-induced changes in gene expression. However, whether these two key proteins act in parallel or as a pathway is not resolved by presented data.

(4) It is very interesting that FABP4 regulates both lipid droplet formation and lipolysis, yet is unclear if the regulation of lipolysis is direct or if the accumulation of lipid droplets - likely plus some other signal(s) - induces upregulation of lipolysis genes.

(5) In several places increased expression of genes coding for enzymes with known functions in lipid biology is conflated with an increase in the lipid biology process the enzymes mediate. Additional evidence would be needed to show these processes are in fact increased in a manner dependent on increased enzyme expression.

---

## [Referee Report · Reviewer #2 (Public review)]

The manuscript by Yorek et al explores the role of fatty acids, particularly unsaturated fatty acids, in lipid droplet accumulation and lipolysis in tumor-associated macrophages (TAMs). Using flow cytometry, immunofluorescent imaging, and TEM, the authors observed that unsaturated fatty acids, such as linoleic acids (LA), tend to induce lipid droplet accumulation in the ER of macrophages, but not in the lysosomes. This phenomenon led them to examine the key enzymes involved in lipid droplet/TAG biosynthesis, where they found incubation of LA upregulates GPAT/DGAT and C/EBPα. In vitro studies, data from public databases, single-cell RNA sequencing of splenic macrophages, and more show that FABP4 emerges as an important mediator for C/EBPα activation. This is further confirmed by FABP4-knockout macrophages, where lipid accumulation and utilization of unsaturated fatty acids were compromised in macrophages through inhibition of LA-induced lipolysis. Using the co-culture system and immunohistochemical analysis, they found that the high FABP4 expression in TAMs, which are observed in metastatic breast cancer tissue, promotes breast cancer cell migration in vitro.

This study is important since the impact of tumor microenvironment is crucial for the development of breast cancer. The individual experiments are well-designed and structured. However, the logic connecting to the next step is a bit difficult to follow, especially when combined with incomplete statistical analysis in some figures, making the conclusion less convincing. For instance, the comparison of macrophage FABP4 expression between breast cancer patients with or without metastasis illustrates the importance of FABP4 expression in metastasis, yet there is no examination of the expression of other key enzymes in the lipolysis or lipid biosynthesis pathway nor there is any correlation with parameters that would reflect patients' consumption of fatty acids. Similarly, an in vivo study comparing FABP4 knockout mice with or without unsaturated fatty acids would yield more compelling evidence. The statistical analysis was largely focused on the sets of unsaturated fatty acids when data from both types of fatty acids were present. In some cases, significant changes are observed in the sets of saturated fat, but there is no explanation of why only the data from unsaturated fats are important for investigation.

Overall, there is solid evidence for the importance of FABP4 expression in TAMs on metastatic breast cancer as well as lipid accumulation by LA in the ER of macrophages. A stronger rationale for the exclusive contribution of unsaturated fatty acids to the utilization of TAMs in breast cancer and a more detailed description and statistical analysis of data will strengthen the findings and resulting claims.

---

## [Referee Report · Reviewer #3 (Public review)]

Summary:

Regulated metabolism has only recently been recognized as a key component of cancer biology, and even more recently recognized as a significant modulator of the tumor microenvironment (TME). TAMs in the TME play a major role in supporting cancer cell survival and growth/spread, as well as generating an immunosuppressive ME to suppress anti-tumor immunity. Specific regulation of lipid metabolism in this context, in particular how lipids are stored and subsequently mobilized for metabolism, is largely unexplored - especially in the immunological components of the TME.

In this manuscript, the authors build on their previous observations that the fatty acid-binding protein FABP4 plays an important role in macrophage function and that FABP4 expression in tumor associated macrophages (TAM) promotes breast cancer progression. They demonstrate:

(1) Unlike saturated fatty acids (FA), unsaturated FA promotes lipid droplet (LD) accumulation in murine macrophages. LD is the primary intracellular storage depot for FA.

(2) Unsaturated FA activates the FABP4-C/EBPalpha axis to upregulate transcription of the enzymes involved in the synthesis of neutral triacylglycerol (TAG) is an essential step in the formation of the neutral lipid core of LD. It should be noted that the authors speculate that UFA-activated FABP4 translocates to the nucleus to activate PPARgamma, which is known to induce C/EBPalpha expression, but do not directly test the involvement of PPARgamma in this axis.

(3) FABP4 deficiency compromises unsaturated FA-mediated lipid accumulation and utilization in murine macrophages.

(4) FABP4-mediated lipid metabolism in macrophages (TAMS) contributes to breast cancer metastasis, in in-vitro of tumor migration induced by murine macrophages and in correlative studies from human patient breast cancer biopsies.

From these studies, the manuscript concludes that FABP4 plays a pivotal role in mediating lipid droplet formation and lipolysis in TAM, which provides lipids to breast cancer cells that contribute to their growth and metastasis.

These are significant findings, as they provide new insight into the mechanistic regulation of TAM biology via regulation of lipid metabolism, as well as define new biomarkers and potential novel therapeutic targets.

The findings are strong in the studies that mechanistically define the role of FAB4 in lipid accumulation and utilization in murine macrophages. However, evidence is less compelling regarding TAM biology and human breast cancer in 3 main areas:

First, while there is clear in vitro evidence that co-cultured murine macrophages genetically deficient in FABP4 (or their conditioned media) do not enhance breast cancer cell motility and invasion, these macrophages are not bonafide TAM - which may have different biology. The use of actual TAM in these experiments would be more compelling. Perhaps more importantly, there is no in vivo data in tumor-bearing mice that macrophage deficiency of FABP4 affects tumor growth or metastasis - which are doable experiments given the availability of the FABP4 KO mice.

Second, no data is presented that the mechanisms/biology that are elegantly demonstrated in the murine macrophages also occur in human macrophages - which would be foundational to translating these findings into human breast cancer. It seems like straightforward in vitro studies in human monocytes/macrophages could be done to recapitulate the main characteristics seen in the murine macrophages.

Third, while the data from the human breast cancer specimens is very intriguing, it is difficult to ascertain how accurate IHC is in determining that the CD163+ cells (TAM) are in fact the same cells expressing FABP4 - which is the central premise of these studies. Demonstrating that IHC has the resolution to do this would be important. Additionally, the in vitro characterization of FABP4 expression in human macrophages would also add strength to these findings.

In summary, the strengths of this manuscript are the significance of metabolic regulation of the immune tumor microenvironment (TME), and the careful mechanistic delineation of FABP4 involvement in mediating lipid droplet formation and lipolysis in murine macrophages. The weaknesses of the work are the lack of direct experimental evidence that human macrophages behave in the same way as murine macrophages, the incomplete characterization of the role of FABP4 expression in TAM in modulating tumor growth in vivo (in murine models), and whether it can be definitively determined that FABP4 is being primarily expressed in the CD163+ macrophages in human breast cancer samples.

Strengths:

(1) Regulated metabolism has only recently been recognized as a key component of cancer biology, and even more recently recognized as a significant modulator of the tumor microenvironment (TME). TAMs in the TME play a major role in supporting cancer cell survival and growth/spread, as well as generating an immunosuppressive ME to suppress anti-tumor immunity.

(2) Regulation of lipid metabolism in this context is largely unexplored, especially in the immunological components of the TME.

(3) The work builds on the authors' previous work on the role of FABP4 plays an important role in macrophage function including FABP4 expression in TAM promotes breast cancer progression (Hao et al, Cancer Res 2018). This paper identified FABP4-expressing macrophages as being pro-tumorigenic via upregulation of IL-6/STAT3 signaling.

(4) The careful and thorough mechanistic delineation of FABP4 involvement in mediating lipid droplet formation and lipolysis in murine macrophages.

(5) The intriguing observations that FABP4-mediated lipid metabolism in macrophages contributes to breast cancer metastasis, in in vitro of tumor migration induced by murine macrophages and in correlative studies from human patient breast cancer biopsies that CD163+ cell numbers (putatively TAM) and FABP4 expression was associated with increased metastatic disease and poor overall survival.

(6) Identification of FABP4 both a prognostic biomarker and a potential therapeutic target to modulate the pro-tumor immune TME.

Weaknesses:

(1) While the authors speculate that UFA-activated FABP4 translocates to the nucleus to activate PPARgamma, which is known to induce C/EBPalpha expression, they do not directly test involvement of PPARgamma in this axis.

(2) While there is clear in vitro evidence that co-cultured murine macrophages genetically deficient in FABP4 (or their conditioned media) do not enhance breast cancer cell motility and invasion, these macrophages are not bonafide TAM - which may have different biology. Use of actual TAM in these experiments would be more compelling. Perhaps more importantly, there is no in vivo data in tumor bearing mice that macrophage-deficiency of FABP4 affects tumor growth or metastasis.

(3) Related to this, the authors find FABP4 in the media and propose that macrophage secreted FABP4 is mediating the tumor migration - but don't do antibody neutralizing experiments to directly demonstrate this.

(4) No data is presented that the mechanisms/biology that are elegantly demonstrated in the murine macrophages also occurs in human macrophages - which would be foundational to translating these findings into human breast cancer.

(5) While the data from the human breast cancer specimens is very intriguing, it is difficult to ascertain how accurate IHC is in determining that the CD163+ cells (TAM) are in fact the same cells expressing FABP4 - which is central premise of these studies. Demonstration that IHC has the resolution to do this would be important. Additionally, the in vitro characterization of FABP4 expression in human macrophages would also add strength to these findings.

---

## [Author Response]

**Reviewer #1:**
(1) After Figure 1, a single saturated (palmitic acid; PA) and a single unsaturated (linoleic acid; LA) fatty acid are used for the remaining studies, bringing into question whether effects are in fact the result of a difference in saturation vs. other potential differences.

PA, SA, OA and LA are the most common FA species in humans (Figure 1A in manuscript). Among them, PA predominantly represents saturated FAs while LA is the main unsaturated FAs, respectively. Of note, although both SA and OA were included in our studies, their effects were comparable to those of PA and LA, respectively. Due to space constraints, the data of SA and OA are not presented in the figures.

(2) While primary macrophages are used in several mechanistic studies, tumor-associated macrophages (TAMs) are not used. Rather, correlative evidence is provided to connect mechanistic studies in macrophage cell lines and primary macrophages to TAMs.

The roe of FABP4 in TAMs has been demonstrated in our previous studies using in vivo animal models1. Therefore, we did not include TAM-specific data in the current study.

(3) CEBPA and FABP4 clearly regulate LA-induced changes in gene expression. However, whether these two key proteins act in parallel or as a pathway is not resolved by presented data.

Multiple lines of evidence in our studies suggest that FABP4 and CEBPA act as a pathway in LA-induced changes: (1) FABP4-negative macrophages exhibit reduced expression of CEBPA in single cell sequencing data; (2) FABP4 KO macrophages exhibited reduced CEBPA expression; (3) LA-induced CEBPA expression in macrophages was compromised when FABP4 was absent.

(4) It is very interesting that FABP4 regulates both lipid droplet formation and lipolysis, yet is unclear if the regulation of lipolysis is direct or if the accumulation of lipid droplets - likely plus some other signal(s) - induces upregulation of lipolysis genes.

Yes, it is likely that tumor cells induce lipolysis signals. Multiple studies have shown that various tumor types stimulate lipolysis to support their growth and progression2-4. In this process, lipid-loaded macrophages have emerged as a promising therapeutic target in cancer5, 6. Consistent with findings that lipolysis is essential for tumor-promoting M2 alternative macrophage activation7, our data using FABP4 WT and KO macrophages demonstrate that FABP4 plays a critical role in LA-induced lipid accumulation and lipolysis for tumor metastasis.

(5) In several places increased expression of genes coding for enzymes with known functions in lipid biology is conflated with an increase in the lipid biology process the enzymes mediate. Additional evidence would be needed to show these processes are in fact increased in a manner dependent on increased enzyme expression.

We fully agree with the reviewer that increased gene expression does not necessarily equate to increased activity. The key finding of this study is that FABP4 plays a pivotal role in linoleic acid (LA)-mediated lipid accumulation and lipolysis in macrophages that promote tumor metastasis. Numerous lipid metabolism-related genes, including FABP4, CEBPA, GPATs, DGATs, and HSL, are involved in this process. While it was not feasible to verify the activity of all these genes, we confirmed the functional roles of key genes like FABP4 and CEBPA through various functional assays, such as gene silencing, knockout cell lines, lipid droplet formation, and tumor migration assays. Supported by established lipid metabolism pathways, our data provide compelling evidence that FABP4 functions as a crucial lipid messenger, facilitating unsaturated fatty acid-driven lipid accumulation and lipolysis in tumor-associated macrophages (TAMs), thus promoting breast cancer metastasis.

**Reviewer #2:**
Overall, there is solid evidence for the importance of FABP4 expression in TAMs on metastatic breast cancer as well as lipid accumulation by LA in the ER of macrophages. A stronger rationale for the exclusive contribution of unsaturated fatty acids to the utilization of TAMs in breast cancer and a more detailed description and statistical analysis of data will strengthen the findings and resulting claims.

We greatly appreciated the positive comments from Reviewer #2. In our study, we evaluated the effects of both saturated and unsaturated fatty acids (FA) on lipid metabolism in macrophages. Our results showed that unsaturated FAs exhibited a preference for lipid accumulation in macrophages compared to saturated FAs. Further analysis revealed that unsaturated LA, but not saturated PA, induced FABP4 nuclear translocation and CEBPA activation, driving the TAG synthesis pathway. For in vitro experiments, statistical analyses were performed using a two-tailed, unpaired student t-test, two-way ANOVA followed by Bonferroni’s multiple comparison test, with GraphPad Prism 9. For experiments analyzing associations of FABP4, TAMs and other factors in breast cancer patients, the Kruskal-Wallis test was applied to compare differences across levels of categorical predictor variable. Additionally, multiple linear regression models were used to examine the association between the predictor variables and outcomes, with log transformation and Box Cox transformation applied to meet the normality assumptions of the model. It is worth noting that in some experiments, only significant differences were observed in groups treated with unsaturated fatty acids. Non-significant results from groups treated with saturated fatty acids were not included in the figures.

**Reviewer #3**
(1) While the authors speculate that UFA-activated FABP4 translocates to the nucleus to activate PPARgamma, which is known to induce C/EBPalpha expression, they do not directly test involvement of PPARgamma in this axis.

Yes, LA induced FABP4 nuclear translocation and activation of PPARgamma in macrophages (see Figure below). Since these findings have been reported in multiple other studies 8, 9, we did not include the data in the current manuscript.

**Author response image 1. sa4fig1:** LA induced PPARg expression in macrophages. Bone-marrow derived macrophages were treated with 400μM saturated FA (SFA), unsaturated FA (UFA) or BSA control for 6 hours. PPARg expression was measured by qPCR (***p<0.001).

(2) While there is clear in vitro evidence that co-cultured murine macrophages genetically deficient in FABP4 (or their conditioned media) do not enhance breast cancer cell motility and invasion, these macrophages are not bonafide TAM - which may have different biology. Use of actual TAM in these experiments would be more compelling. Perhaps more importantly, there is no in vivo data in tumor bearing mice that macrophage-deficiency of FABP4 affects tumor growth or metastasis.

In our previous studies, we have shown that macrophage-deficiency of FABP4 reduced tumor growth and metastasis in vivo in mouse models1.

(3) Related to this, the authors find FABP4 in the media and propose that macrophage secreted FABP4 is mediating the tumor migration - but don't do antibody neutralizing experiments to directly demonstrate this.

Yes, we have recently published a paper of developing anti-FABP4 antibody for treatment of breast cancer in moue models10.

(4) No data is presented that the mechanisms/biology that are elegantly demonstrated in the murine macrophages also occurs in human macrophages - which would be foundational to translating these findings into human breast cancer.

Thanks for the excellent suggestions. Since this manuscript primarily focuses on mechanistic studies using mouse models, we plan to apply these findings in our future human studies.

(5) While the data from the human breast cancer specimens is very intriguing, it is difficult to ascertain how accurate IHC is in determining that the CD163+ cells (TAM) are in fact the same cells expressing FABP4 - which is central premise of these studies. Demonstration that IHC has the resolution to do this would be important. Additionally, the in vitro characterization of FABP4 expression in human macrophages would also add strength to these findings.

The expression of FABP4 in CD163+ TAM observed through IHC is consistent with our previous findings, where we confirmed FABP4 expression in CD163+ TAMs using confocal microscopy. Emerging evidence further supports the pro-tumor role of FABP4 expression in human macrophages across various types of obesity-associated cancers11-13.

References

(1) Hao J, Yan F, Zhang Y, Triplett A, Zhang Y, Schultz DA, Sun Y, Zeng J, Silverstein KAT, Zheng Q, Bernlohr DA, Cleary MP, Egilmez NK, Sauter E, Liu S, Suttles J, Li B. Expression of Adipocyte/Macrophage Fatty Acid-Binding Protein in Tumor-Associated Macrophages Promotes Breast Cancer Progression. Cancer Res. 2018;78(9):2343-55. Epub 2018/02/14. doi: 10.1158/0008-5472.CAN-17-2465. PubMed PMID: 29437708; PMCID: PMC5932212.

(2) Nieman KM, Kenny HA, Penicka CV, Ladanyi A, Buell-Gutbrod R, Zillhardt MR, Romero IL, Carey MS, Mills GB, Hotamisligil GS, Yamada SD, Peter ME, Gwin K, Lengyel E. Adipocytes promote ovarian cancer metastasis and provide energy for rapid tumor growth. Nat Med. 2011;17(11):1498-503. Epub 20111030. doi: 10.1038/nm.2492. PubMed PMID: 22037646; PMCID: PMC4157349.

(3) Wang YY, Attane C, Milhas D, Dirat B, Dauvillier S, Guerard A, Gilhodes J, Lazar I, Alet N, Laurent V, Le Gonidec S, Biard D, Herve C, Bost F, Ren GS, Bono F, Escourrou G, Prentki M, Nieto L, Valet P, Muller C. Mammary adipocytes stimulate breast cancer invasion through metabolic remodeling of tumor cells. JCI Insight. 2017;2(4):e87489. Epub 20170223. doi: 10.1172/jci.insight.87489. PubMed PMID: 28239646; PMCID: PMC5313068.

(4) Balaban S, Shearer RF, Lee LS, van Geldermalsen M, Schreuder M, Shtein HC, Cairns R, Thomas KC, Fazakerley DJ, Grewal T, Holst J, Saunders DN, Hoy AJ. Adipocyte lipolysis links obesity to breast cancer growth: adipocyte-derived fatty acids drive breast cancer cell proliferation and migration. Cancer Metab. 2017;5:1. Epub 20170113. doi: 10.1186/s40170-016-0163-7. PubMed PMID: 28101337; PMCID: PMC5237166.

(5) Masetti M, Carriero R, Portale F, Marelli G, Morina N, Pandini M, Iovino M, Partini B, Erreni M, Ponzetta A, Magrini E, Colombo P, Elefante G, Colombo FS, den Haan JMM, Peano C, Cibella J, Termanini A, Kunderfranco P, Brummelman J, Chung MWH, Lazzeri M, Hurle R, Casale P, Lugli E, DePinho RA, Mukhopadhyay S, Gordon S, Di Mitri D. Lipid-loaded tumor-associated macrophages sustain tumor growth and invasiveness in prostate cancer. J Exp Med. 2022;219(2). Epub 20211217. doi: 10.1084/jem.20210564. PubMed PMID: 34919143; PMCID: PMC8932635.

(6) Marelli G, Morina N, Portale F, Pandini M, Iovino M, Di Conza G, Ho PC, Di Mitri D. Lipid-loaded macrophages as new therapeutic target in cancer. J Immunother Cancer. 2022;10(7). doi: 10.1136/jitc-2022-004584. PubMed PMID: 35798535; PMCID: PMC9263925.

(7) Huang SC, Everts B, Ivanova Y, O'Sullivan D, Nascimento M, Smith AM, Beatty W, Love-Gregory L, Lam WY, O'Neill CM, Yan C, Du H, Abumrad NA, Urban JF, Jr., Artyomov MN, Pearce EL, Pearce EJ. Cell-intrinsic lysosomal lipolysis is essential for alternative activation of macrophages. Nat Immunol. 2014;15(9):846-55. Epub 2014/08/05. doi: 10.1038/ni.2956. PubMed PMID: 25086775; PMCID: PMC4139419.

(8) Gillilan RE, Ayers SD, Noy N. Structural basis for activation of fatty acid-binding protein 4. J Mol Biol. 2007;372(5):1246-60. Epub 2007/09/01. doi: 10.1016/j.jmb.2007.07.040. PubMed PMID: 17761196; PMCID: PMC2032018.

(9) Bassaganya-Riera J, Reynolds K, Martino-Catt S, Cui Y, Hennighausen L, Gonzalez F, Rohrer J, Benninghoff AU, Hontecillas R. Activation of PPAR gamma and delta by conjugated linoleic acid mediates protection from experimental inflammatory bowel disease. Gastroenterology. 2004;127(3):777-91. doi: 10.1053/j.gastro.2004.06.049. PubMed PMID: 15362034.

(10) Hao J, Jin R, Yi Y, Jiang X, Yu J, Xu Z, Schnicker NJ, Chimenti MS, Sugg SL, Li B. Development of a humanized anti-FABP4 monoclonal antibody for potential treatment of breast cancer. Breast Cancer Res. 2024;26(1):119. Epub 20240725. doi: 10.1186/s13058-024-01873-y. PubMed PMID: 39054536; PMCID: PMC11270797.

(11) Liu S, Wu D, Fan Z, Yang J, Li Y, Meng Y, Gao C, Zhan H. FABP4 in obesity-associated carcinogenesis: Novel insights into mechanisms and therapeutic implications. Front Mol Biosci. 2022;9:973955. Epub 20220819. doi: 10.3389/fmolb.2022.973955. PubMed PMID: 36060264; PMCID: PMC9438896.

(12) Miao L, Zhuo Z, Tang J, Huang X, Liu J, Wang HY, Xia H, He J. FABP4 deactivates NF-kappaB-IL1alpha pathway by ubiquitinating ATPB in tumor-associated macrophages and promotes neuroblastoma progression. Clin Transl Med. 2021;11(4):e395. doi: 10.1002/ctm2.395. PubMed PMID: 33931964; PMCID: PMC8087928.

(13) Yang J, Liu S, Li Y, Fan Z, Meng Y, Zhou B, Zhang G, Zhan H. FABP4 in macrophages facilitates obesity-associated pancreatic cancer progression via the NLRP3/IL-1beta axis. Cancer Lett. 2023;575:216403. Epub 20230921. doi: 10.1016/j.canlet.2023.216403. PubMed PMID: 37741433.